**Small- and meso-scale field-aligned auroral current structures, their spatial and temporal**
**characteristics deduced by Swarm constellation**
Hermann Lühr [1*] and Yun-Liang Zhou [2]
1. GFZ Helmholtz Centre for Geosciences, Section 2.3, Geomagnetism, Telegrafenberg, 14473
Potsdam, Germany
2. School of Earth and Space Science and Technology, Wuhan University, 430072 Wuhan,
China
*Correspondence to: hluehr@gfz.de
**Abstract** Magnetic field recordings by the Swarm A and C spacecraft during the Counter Rotation
Orbit phase are used for checking the stationarity of auroral region small- and meso-scale field-
aligned currents (FAC). The varying separations between the spacecraft in along- and cross-track
direction during this constellation phase allow for determining the spatial and temporal correlation
lengths for FAC structures of different along-track wavelengths. We make use of the cross-
correlation analysis to check the agreement of the magnetic signatures at the two spacecraft. When
the cross-correlation coefficient exceeds 0.75 at a time lag that equals the along-track time
difference, the event is identified as stationary. It is found that meso-scale FACs of along-track
wavelengths >100 km are primarily stable for more than 40 s and over cross-track separations
exceeding 20 km. A prime reason for their occasional deselection is the latitudinal motion of the
current system. Conversely, stationary small-scale FACs (10-50 km scale sizes) are found to have
typically azimuthal sizes of only 12 km. Their temporal stability is limited to about 18 s. This class
of small-scale FACs is commonly found in the cusp region and prenoon sector at 75°- 80° magnetic
latitude. In the midnight sector these FACs are weaker and appears at lower latitudes. The small-
scale FACs are assumed to be associated with kinetic Alfvén wave. There exists another class of
kilometer-scale FACs, which cannot be well resolve by our dataset, but seem to influence our
analysis.

## 1. Introduction

Field-aligned currents (FAC) are commonly observed in magnetized plasmas due to the high
electric conductivity along magnetic field lines. In near-Earth space they can be found in the
ionosphere and magnetosphere, with particularly strong currents flowing within the auroral
regions. FACs appear at a wide range of horizontal scales from less than a kilometer (e.g., Neubert
and Christiansen, 2003; Rother et al., 2007) up to 1000 km (e.g., Iijima and Potemra, 1976;
Anderson et al., 2014). Commonly, the observed current densities become larger the smaller the
scales are. Pairs of upward and downward currents with comparable scales are mostly close
together. An important generation region for FACs on the dayside is the low-latitude boundary
layer (LLBL) (e.g., Siscoe, 1991). As outlined by Johnson and Wing (2015); Wing and Johnson
(2015) the flow shear between the plasma in the magnetosheath and the almost stagnant plasma in
the magnetosphere is the main driver of the currents flowing along the field lines into and out of
the ionosphere. Since the current density is depending on the properties of the LLBL, variations
in its thickness and plasma density will cause meso-scale FAC features, with larger amplitudes,
interleaved into the large-scale FAC regimes.
Besides the large-scale FACs, flowing practically all the time, there exist transient FAC features.
Typical source mechanisms for them are, e.g., Kelvin-Helmholtz vortices at the flanks (Johnson
et al. 2021; Petrinec et al., 2022), flux transfer events (Omidi and Sibeck, 2007; Ala-Lahti et al.,
2022), or travelling convection vortices (Friis-Christensen et al., 1988; Lühr et al., 1998). All these
processes can drive FACs of meso to small scales in the ionosphere (some 100 km or less).
Large-scale FACs at auroral latitudes can be treated as quasi-static circuits, having their source in
the magnetosphere, and the closure current in the ionosphere acts as a dissipative load. For smaller
scales of FACs, approaching some 10s of km, the dynamic characteristics of the circuit become
important, and reflection at the ionosphere plays a role (e.g. Lysak, 1990; Vogt and Haerendel,
1998). Thus, Alfvén waves, carried by FACs, are becoming the dominant feature in the 10-50 km
scale range. Ishii et al. (1992) made use of magnetic ($B$) and electric field ($E$) measurements by
the Dynamic Explorer 2 satellite for distinguishing between waves and static current circuits. From
the amplitude relation between the two fields, $E/\Delta B$, they could discriminate the two types of FAC
observations, showing that a cutoff exists for stationary FACs at the small-periods end around 4 s
to 10 s. When considering the satellite velocity of about 7.5 km/s, the apparent periods convert to
latitudinal wavelengths of 30 km to 75 km. Similarly, Pakhotin et al. (2018) made use of Swarm
satellite constellation data and investigated FAC characteristics by a comparison between electric
field and magnetic field data for one auroral region crossing. They report for their case a change
of current characteristics around a period of 5 s (~40 km wavelength) from quasi-static to dynamic,
which is well in line with the results of Ishii et al. (1992).
When deriving FAC estimates from satellites, one important assumption is the stationary of the
magnetic signal over the time of measurement. This is not always satisfied. As reported by
Stasiewicz et al. (2000) in their review article, small-scale FAC structures are commonly
associated with kinetic Alfvén waves. Therefore, the current strength exhibits a significant
temporal variation. From single-satellite magnetic field measurements it is not possible to
distinguish between temporal and spatial variations. To overcome this problem Gjerloev et al.
(2011) made use of the three ST5 satellites in pearls-on-a-string formation. They performed a large
statistical study of the FAC temporal stability depending on their along-track scale size. On the
nightside FAC structures larger than 100 km had been found to be quasi-stationary, while on the
dayside this was only true for scales above 200 km. Due to their orbital geometry, the ST5
spacecraft were cycling the Earth in a sun-synchronous mode, which provided only little local time
coverage. Furthermore, the orbits of the three spacecraft were well lined up. Therefore, no
information on the longitudinal correlation length of FAC structures could be achieved. A similar
study of FAC spatial and temporal scales was performed by Lühr et al. (2015), making use of the
three Swarm spacecraft soon after launch when the satellites where slowly separating from each
other. These authors generally confirmed the results of Gjerloev et al. (2011). FACs of scales up
to some 10s of kilometer are highly variable with typical persistent periods of about 10 s. Larger-
scale FACs (>150 km) can be regarded as quasi-stationary, being stable over more than 60 s. The
longitudinal extent of the small FAC sheets was reported to be about 4 times large on the nightside
than their latitudinal scale, but on the dayside both scales were found to be of comparable size. In
spite of these valuable results, the study by Lühr et al. (2015) had a number of limitations. The
data were taken during December 2013 - January 2014, over less than 50 days. Thereafter,
dedicated orbit maneuvers were performed. This means, it was not possible to investigate any
seasonal dependences nor get a good local time coverage.
One of the standard Swarm Level-2 data products is the FAC density estimate (Ritter et al, 2013;
Lühr et al., 2020) derived from the magnetic field measurements of the Swarm A and Swarm C
spacecraft flying almost side-by-side with only a small along-track separation of around 7 s. An
important assumption for the reliability of the product is that both satellites record magnetic field
variation caused by the same FAC structure. In a dedicated study Forsyth et al. (2017) compared
the recordings at the two Swarm spacecraft by means of a cross-correlation analysis. They

generally find large cross-correlation coefficients when the magnetic field data are low-pass filtered with a cutoff of about 20 s. In addition, these authors request a similarity in signal amplitude at the two satellites, not exceeding a difference of 10%. This request significantly reduces the number of suitable data pairs for FAC estimates. We do not consider their amplitude request as justified because the FAC estimate derived from the dual-spacecraft approach is the mean value of the current density passing through the integration loop at measurement time. Thus, the dominating linear parts of the spatial and temporal gradients are taken care of by the integration.

What is missing after all these studies, is a detailed investigation of the smaller-scale FACs. What are their spatial, temporal correlation lengths? This information is needed, e.g., for determining the range over which dual-spacecraft FAC estimates are valid. An opportunity for this kind FAC scale-size analysis arose during the Swarm Counter-Rotating Orbit Phase in 2021. During that campaign the orbits of Swarm A and C were brought close together, and Swarm B cycled the Earth in opposite direction. Early October 2021 all three orbital planes were quasi-coplanar. Thereafter, the orbits slowly separated again. For the study presented here we make use of the magnetic field recordings from 1 May 2021 to 28 February 2022. The dataset covers both June and December solstices and for both seasons all local times are visited. We make use of the cross-correlation analysis, which is applied to the transverse magnetic field component of the Swarm A and C satellites. As a result, we obtain the variation of cross-correlation coefficient and its dependence on the spacecraft separations in along- and cross-track directions. This allows us to address a number of remaining questions concerning the temporal and spatial scales of small- and meso-scale FAC structures. Particular attention is paid in this study to FAC structures with apparent periods of 3 to 15 s corresponding to along-track wavelengths of about 20 to 100 km.

In the section to follow we present the Swarm mission and magnetic field data, in addition, the preprocessing is described. Section 3 introduces the cross-correlation analysis, our prime tool for determining the similarity between the magnetic field signatures at the two spacecraft. This is followed by a statistical analysis, in Section 4, of the derived correlation results during the 10 months of our study period. It includes both the characteristics of the identified stable and unstable FAC events separately for 6 spatial scale ranges. The obtained results are discussed in Section 5. Here we try to explain the behavior of the current structures and compare the findings with earlier publications. The Summary and Conclusions give a wrap-up of the derived auroral zone FAC stationarity characteristics.

## 2. Data and processing approach

ESA's Swarm satellite mission was launched on 22 November 2013. It comprises three identical satellites in near-polar orbits at different altitudes (Friis-Christensen et al., 2008). During the first mission phase, starting on 17 April 2014, the lower pair, Swarm A and C, was flying at an altitude of about 450 km almost side-by-side with a longitudinal separation of 1.4° and with Swarm C being lagged along-track by about 7 s; while the third, Swarm B, cruised about 60 km higher. Due to their orbital inclination of 87.3°, Swarm A/C need about 133 days to cover all 24 local time (LT) hours, when considering both ascending and descending orbital arcs. Swarm B, with a slightly larger inclination (88°) needs about 141 days for a full local time coverage. As a consequence of the difference in inclination, the angle between the orbital planes of Swarm A/C and Swarm B slowly increased.

In preparation for a second mission phase, the counter-rotating orbits campaign, the longitudinal separation between Swarm A and C was slowly reduced starting in October 2019. Around 1 October 2021 the orbital planes of Swarm A/C and Swarm B were quasi coplanar. All three crossed the equator at similar longitudes, with Swarm B flying in opposite direction (e.g., Xiong and Lühr, 2023). Thereafter, the separation between Swarm A and C orbital planes increased again at a rate

of 0.7° in longitude per year (for more details about the constellation evolution see Fig. 1 in Zhou
et al., 2024). The months of small Swarm A/C separation, around the epoch of coplanarity, are of
special interest for this study.
Here we make use of the Swarm Level-1b 1 Hz magnetic field data with product identifier
"MAGx_LR", where lower-case "x" in the product name represents a placeholder for the
spacecraft, A, B, C. The magnetic vector data are given in the North-East-Center (NEC) frame.
For our purpose we remove from the Swarm A and C magnetic field observations the contributions
of core, crustal, and magnetospheric fields by subtracting the geomagnetic field model CHAOS-
7.11 (Finlay et al., 2020). The residuals of the horizontal components, $Bx$ and $By$, are used for
studying the magnetic signatures caused by the FACs. The limitation to these two components is
justified at auroral latitudes since the magnetic field lines are almost vertical to the Earth surface.
From these two components we calculate the deflections of, $B_{trans}$, transverse to the flight
direction.

$$B_{trans} = B_y \, cos(\gamma) - B_x \, sin(\gamma) \qquad (1)$$

where $sin(\gamma) = cos(incl)/cos(lat)$, with $incl$ as orbital inclination and $lat$ as latitude at the
measurement point. For application in Eq. (1) $\gamma = \gamma$ has to be used on the ascending part of the
orbit and $\gamma = \pi - \gamma$ on the descending part.
We actually used in our further investigations the difference between two adjacent values of $B_{trans}$,
separated by 1 s. These differences, $\Delta B_{trans}$, help to remove remaining large-scale biases after
model subtraction; furthermore, they better represent the characteristics of FACs. A related single-
satellite FAC estimate would read

$$j_z = \frac{1}{\mu_0} \frac{\Delta B_{trans}}{v_{SC}} \qquad (2)$$

where $\mu_0$ is the vacuum permeability and $v_{SC}$ is the spacecraft velocity. When inserting, for
example, $\Delta B_{trans} = 10$ nT/s and the typical orbital speed of Swarm, 7.5 km/s, we obtain for the
FAC density, $j_z = 1.1 \, \mu A/m^2$. This means, the FAC densities amount approximately to a tenth of
the B-field change rate.
Several phenomena are associated with FACs of different scale sizes. In order to identify the scale-
dependent properties we subdivide the signal of $\Delta B_{trans}$ into six quasi-logarithmically spaced
period bands. The chosen -3 dB pass-band filter limits are 1-3 s, 3-7 s, 7-13 s, 13-23 s, 23-39 s,
and 39-60 s. For the estimation of FACs it is generally assumed that the magnetic field variations
are caused by the passage through a static structure. Therefore, the above periods can be converted
to wavelength by multiplication with the satellite speed. Within a wavelength both the upward and
downward FACs are included. When talking about the scale size of a FAC, commonly only the up
or down part is meant. In the following we thus use half the wavelength for the scale size of a
FACs. The term small-scale FAC is used in the study for scale sizes of about 10 to 50 km and
FACs with sizes of about 75 to 220 km are termed here meso-scale. In the gap between the two
ranges characteristics of both types may be found.
The studied time interval lasts from 1 May 2021 to 28 February 2022. During these months the
Swarm A/C cross-track separation is small and stays below 20 km at auroral latitudes. The along-
track separation between the spacecraft is deliberately modified around the time of coplanarity
between 2 s and 41 s, as shown in Figure 1 of Zhou et al. (2024). The same figure displays also
the variation of longitudinal (cross-track) separation at the equator over the study period. Here we
are interested in the small and meso-scale FAC characteristics at auroral latitudes beyond ±60°
magnetic latitude (MLat).

## 3. Correlation analysis

A common assumption, when estimating FAC density from magnetic field satellite observations, is that the signal is quasi-stationary. This condition can be tested, when Swarm A and C observe the same current structure. We check the stationarity with the help of a cross-correlation analysis.

$$Cc = \frac{\sum[(X-X_m)\cdot(Y-Y_m)]}{\sqrt{\sum(X-X_m)^2\cdot\sum(Y-Y_m)^2}} \tag{3}$$

where, $X$ represents the signal amplitude of $\Delta B_{trans}$ from Swarm A, $Y$ represents the signal of $\Delta B_{trans}$ from Swarm C, and $Xm$ and $Ym$ represent the mean values of $\Delta B_{trans}$ over the correlation intervals of the two satellites, respectively. We further determine the maxima of $Cc$ and the corresponding time lags (T-lag) between the two satellite data series. It has to be noted that in this way T-lag is derived only with a one-second resolution, as the Swarm magnetic measurements have a 1-s resolution. For improving this situation, we consider the five cross-correlation results centered on the maximum $Cc$ value and apply a fourth-order polynomial fitting. From the peak location of the fitted polynomial, we get the T-lag in fractions of a second. According to the definition of $Cc$ in Eq. (3), a positive T-lag means, Swarm A samples the magnetic signal before Swarm C.

Figure 1 presents examples of passes over the north pole. The upper frame is from 1 September 2021, along a midnight - noon orbit. For this and the example-pass in the lower frame cross-correlations have been applied to recordings of the two satellites. Here we consider a sliding window of 60 s (corresponding to a distance of 450 km) of $\Delta B_{trans}$ from Swarm A and C for deriving the peak correlation coefficients, $Cc$. In addition, the root mean square (RMS) amplitude of the signal is calculated. The $\Delta B_{trans}$ in the top panel shows two groups of signal bursts at auroral latitudes in the recordings of the two satellites. These represent intense FAC at a wide range of scales, the earlier one occurring around midnight and the other at noon. For getting an impression of the related FAC densities, a $\Delta B_{trans}$ amplitude of 100 nT/s corresponds, as shown above, to approximately 10 $\mu A/m^2$. The RMS values are given in the second panel. In the third panel optimal T-lag values are plotted. Over large parts of the pass it stays close to the actual time shift, $\Delta t = 4$ s, between the spacecraft, as expected for a static structure. The $Cc$ values at the bottom herald good correlation ($Cc > 0.75$) for most of the signal. However, that is not always the case for the bursts. In this example, the equatorward part of the nightside burst exhibits an almost perfect correlation at expected lag time, while the correlation on the poleward side is poor. At noon the cross-correlation coefficient is also below the threshold on the poleward parts but is high during the stronger later burst.

In the lower frame of Figure 1 an example from 1 January 2022 is displayed. The recordings, in the same format as above, are again from a noon - midnight pass. Bursts of small-scale FACs appear around 78° MLat. An obvious difference to the example shown above is the generally low cross-correlation coefficient. Here again, the more equatorward parts of the bursts tend to show larger $Cc$. The major difference of this example compared to the former is the much longer along-track separation between the two satellites of $\Delta t = 22.3$ s. In spite of that enlarged distance, best correlation between the two recordings is achieved at time lags close to the actual time difference, but the quality of correlation has decreased considerably.

These examples show that small-scale FACs do reach large amplitudes at auroral latitudes, but they cannot generally be considered as static structures. For that reason, we decided to perform the cross-correlation analysis separately for all the six period bands and draw conclusions about the correlation length depending on the FAC scale sizes. For the various cross-correlations the interval lengths and step sizes are listed in Table 1.

**Table 1:** Listing of the data interval lengths and step sizes for the cross-correlation analysis of the various period bands

| Period band | Scale-length | Data interval | Step size |
|:---:|:---:|:---:|:---:|
| **1 -3 s** | 4-11 km | 10 s | 2 s |
| **3 - 7 s** | 11-26 km | 20 s | 5 s |
| **7 - 13 s** | 26-49 km | 40 s | 10 s |
| **13 - 23 s** | 49-86 km | 70 s | 17 s |
| **23 - 39 s** | 86-156 km | 120 s | 30 s |
| **39 - 60 s** | 156-225 km | 180 s | 45 s |

## 4. Statistical analysis

The aim of our study is to identify the typical properties of small- and meso-scale FAC structures at auroral latitudes. Such information can best be achieved when simultaneous observations at multiple points are available. For the statistical analysis we considered the time 1 May 2021 - 28 February 2022, when the orbital planes of Swarm A and C differed only by small angles (<0.3°). This promises meaningful multipoint measurements down to the smallest resolvable scales of <10 km.

The study interval overlaps with the trailing part of the solar minimum. Therefore, the mean solar flux level increases slowly from F10.7 = 80 sfu in the beginning to about F10.7 = 110 sfu around the end. The magnetic activity stays generally below Kp = 4. There were just some stormy days, quite outstanding is 4 November 2021, followed at decreasing activity levels by 12 October 2021 and 4 February 2022. Due to the generally calm conditions, the derived mean FAC characteristics represent low to normal solar wind driving states.

It is known from earlier publications that FACs of different scales exhibit different dynamic properties, e.g. Stasiewicz et al. (2000) and references therein. For that reason, we divided the magnetic variations $\Delta B_{trans}$ into the six period bands as outlined in Section 2 and Table 1. The cross-correlation analysis, as defined in Eq. (3), was applied separately to all the period bands and all the days. A current structure is considered as stationary when the magnetic field measurements from the two Swarm spacecraft achieve a peak cross-correlation coefficient, $Cc > 0.75$, at a time shift that fits the along-track separation, $\Delta t \pm 1.5$ s. In order to avoid fault results from too small signals, in addition a minimum RMS amplitude is requested. For the choice of suitable thresholds, we considered the following facts. In their study on low-latitude FAC properties Zhou et al. (2024) had chosen a rather small amplitude limit, RMS > 0.03 nT/s. As can be seen from their Figure 2, magnetic field variations of FAC signatures at mid-latitude frequently surpass 1 nT/s. Those FACs are generally driven by atmospheric processes, as winds and waves. Here we are interested in FACs driven by magnetospheric processes, therefore those from atmospheric dynamics, also active at auroral latitudes, should be suppressed. For that reason, we decided to choose a threshold, RMS > 2 nT/s, in this study. For assessing the consequence of this selection, Figure 2 shows the occurrence distribution of wavy magnetic signals at auroral latitudes beyond 60° MLat and within the period range of 1-60 s (corresponding to along-track wavelengths up to 450 km). As expected, the occurrence rate declines, the larger the threshold. For RMS > 2 nT/s about 60% of the satellite recordings provide significant FAC signal. This percentage fits also approximately the typical fraction of the auroral oval latitude range that is covered by the Iijima and Potemra (1976) type FACs.


*4.1 Variation of stationary structures over the study period*


Here we start with a look at the variable magnetic field signal that could be caused by FACs and
how this activity varies over the cause of our study period. Figure 3 presents the occurrence
frequency, on a daily basis, of wavy signals with amplitudes RMS > 2 nT/s, as recorded by Swarm
A, separately for the six period ranges and the two hemispheres. No comparison with Swarm C
measurements has been performed. Across the bottom of the figure additional information is
added, the typical orbital local time (taken at 70° MLat) separately for up- and downleg arcs, the
along-track time difference, $\Delta t$, between Swarm A and C, and the east-west separation of the
spacecraft at the equator, $\Delta$Long, in degree.
The curves for the different periods are quite similar. They reach up to rates of 50%. This is
expected as a result of our chosen RMS threshold. General features to be noticed; higher rates are
observed at local summer conditions. This is the case during the first half of the study period in
the northern hemisphere and the second half in the southern. More events are found during daytime
than at night. This holds for both hemispheres but is more pronounced for shorter periods (smaller
structures) and for local summer time. The occurrence rate varies partly at regular period. This is
close to 13.5 days (see arrow in top panel), reflecting half the rotation period of the sun. There
appears a particularly deep dip in occurrence rate at 27-28 October 2021. This is a time of very
low magnetic activity, having obviously direct influence on the FAC activity at auroral latitudes.
Only one week later, 3-4 November, a major magnetic storm occurred, giving rise to enhanced
signal rates, in particular, in the southern hemisphere (see the arrows pointing at the two events).
For comparison, Figure 4 shows for the two hemispheres the temporal variation of the mean
occurrence rates of positively detected stationary structures by means of cross-correlation between
Swarm A and C recordings, separately for the up- and downleg orbital arcs within auroral latitudes
(beyond 60° MLat), in the same format as before. Particularly low event occurrences are found at
the smallest scale (1-3 s, 4-11 km). Only during the 2 weeks around coplanarity (1 October)
somewhat enhanced rates appear. Within that fortnight the along-track separation was reduced to
$\Delta t = 2$ s. This indicates rather short correlation times for these narrow FACs. For larger current
structures the curves start to agree more and more with the before shown activity curves, at least
for the first half of the study period. During the second half we find a prominent dip in event
occurrences centered on December 2021. In that month the along-track separation is particularly
large between Swarm A and C.
Finally, Figure 5 presents the ratio of magnetic signatures that can be interpreted as stationary FAC
structures. Plotted are the daily detected static structures divided by the counts of activity intervals.
These normalized curves reveal which fraction of the high latitudes is covered by stationary FACs
of different scale-lengths. Rather similar curves are derived from the two hemispheres. Towards
longer periods (larger scales) fairly constant ratios between 80% and 100% are obtained. This
confirms that almost all FACs of these scales at auroral latitudes can be considered as quasi-static.
The rates start to drop for shorter periods shorter than ~10 s (<38 km scales). Particularly low
percentages are returned for the very short period (1-3 s). Also here, the rates are somewhat
outstanding in the 2 weeks around orbit coplanarity, when the spacecraft separation was reduced
to 2 s. Peak occurrence ratios can also be found for other period bands during those two weeks.
When normalizing the identified stationary FAC structures by the number of wavy signal events
with amplitudes above the threshold (RMS > 2 nT/s), the environmental influences on the
occurrence rates of such features, depending possibly on solar wind input, local time or season,
are largely removed, but the effect of the Swarm constellation on positive detections prevails.
Therefore, these plots are more suitable for evaluating the properties of the detected currents.
During the first half of the study period we find the blue curves in Figure 5 on higher levels than

the red in the northern hemisphere. This means larger ratios of stationary FACs are obtained in the afternoon to late evening sector, compared to the early morning to prenoon local times during summer season. Conversely, when looking at the southern hemisphere (right frame), where winter conditions prevail during the first half, the red lines tend to be higher than the blue. This means, a higher percentage of stable FACs in the morning than in the evening sector.

For larger structures, >75 km (>20 s period), the ratios are almost 100% everywhere. This means, practically all magnetic fluctuations lasting longer than 20 s can be regarded as caused by quasi-stable FAC structures. Exceptions appear during November, December 2021. According to Figure 1 of Zhou et al. (2024), during those months the along-track separation between the satellites was above 20 s, increased up to 41 s on 16 December, after which it linearly decreased.

These ratio plots provide the opportunity to distinguish between the temporal and spatial correlation-lengths of the different FAC scales. From Figure 1 of Zhou et al. (2024) it can be seen that the along-track separation does not vary too much about its mean of ~5 s in the beginning of our study period up to mid-September. During these early months the occurrence ratios, for example of the 3-7 s and 7-13 s period structures (corresponding to scales of 10 - 50 km) exhibit in both hemispheres linear increases from the beginning of the study interval up to the date of very shortest along-track separation (18 September). We may attribute this behavior to the change of cross-track separation, and as expected, its influence is stronger the shorter the signal period. Quite differently, during the time after closest approach (past 5 October) we find first a rapid decrease and after 16 December a recovery of the occurrence ratios. This variation of the ratio we like to attribute to the combination of growing along- and cross-track separations. It is worth noting that the ratios for all periods show approximately the same values at the beginning and end of the study time, when along- and cross-track separations have attained again similar values. For obtaining an estimate of the influence of the along-track separation on the ratio, one should first remove the cross-track effect, as observed during the time before coplanarity. Without going into details, it is obvious to see that also the along-track separation has a stronger effect on the shorter periods. The number of commonly observed stationary structures, for example of the 3-7 s period signals, is reduced by this effect to more than a factor of 2 at $\Delta t = 40$ s. Smaller reductions are derived for longer periods (larger scales). These observations provide a first idea of the FAC structure's temporal correlation lengths.

Somewhat special are the occurrence ratios obtained for the 1-3 s periods. Outside the true coplanarity period with $\Delta t = 2$ s the ratios are quite low, varying around 20%. Interestingly, they do not approach zero. This indicates that there exist a few 4-11 km scale FACs that are stationary over the ranges of along- and cross-track separations considered here. We will revisit this issue at the end of Section 5.2.

Another feature, worth to be noted, is the local minimum in FAC ratios on 4 November 2021. This coincides with the strongest magnetic storm during our study period. Suggested reasons for the reduced number of stationary FACs are the more dynamic character of the currents with smaller scales and/or the expansion of the auroral oval beyond our limit of 60° MLat for the larger scales. No such effect on the ratios has been found during the other, weaker storm times. The storm-related reduction of stable FACs on 4 November is present at all scale lengths but is more prominent at shorter scales. Conversely, for the very quiet days, 27-29 October 2021, a dip in ratio appears only at long periods.

An even more detailed view on the occurrence of selected FAC events is provided by Figure 6. It shows the latitudinal distribution of the ratio of stable FAC structures. The prime features of Figure 5 are also visible here, but this presentation offers another dimension. When starting with the first half of the study period, before the time of orbital coplanarity, we find for periods larger than ~20 s at all the latitudes, for all local times and both hemispheres high occurrence ratios. For shorter period, say <15 s, higher ratios tend to concentrate in the northern hemisphere at very polar

latitudes. This could be due to some current properties, but it has to be taken into account that the orbital cross-over takes place close to the pole and with that, small cross-track separations are found there. However, when turning to the southern hemisphere (lower frame of Figure 6) we conversely find, e.g. for the 7-13 s period band, largest ratios at the low-latitude end. This means, our expectation of large ratios near the cross-over is obviously not confirmed by the southern hemisphere observations. Rather, it seems that the season, summer in the norther hemisphere, is responsible for the preferred appearance of stationary small-scale FACs at high latitudes. Conversely, in the southern, winter hemisphere these small-scale high-latitude FACs are missing, as is obvious from the blank parts in the plot near the pole. The effect of cross-track distance on this period band is stronger at lower latitudes. The larger distances between spacecraft cause more reduced occurrence ratios in the morning and prenoon sectors than at afternoon to evening.

The second part of the study period exhibits more dynamic variations. Most prominent is the obvious drop in ratio during the time of extended along-track separation, Δt. As mentioned already above, the reduction is most pronounced for the small periods (short scales) and becomes less prominent the longer the period. On the upleg passes, covering the noon to afternoon sector, the reductions in ratio are quite evenly distributed over the latitudes. Conversely for the downleg, sampling night and early morning hours, there is a clear preference of ratio reduction at higher latitudes. Differently, the lower-latitude auroral FAC structures (e.g. R2 FACs) in this sector seem to be stable for more than 40 s.

Worth mentioning are also the two narrow features of low ratios around the beginning of November 2021. They represent contrasting activity conditions, the quietest period (27-29 October 2021) and the most intense storm on 3-4 November 2021 of the study period. In an attempt to explain the reasons for the low ratios we had a look at the cross-correlation plots from the two very different days. An example of a pass from the quiet day over the southern auroral region is shown in Figure 7, top frame. We find moderate activity in the morning and night sector at fairly high latitudes, around 70° MLat. For most parts of the orbital arc high correlation coefficients are obtained at the right time lag. Just the burst at night-time makes an exception. In spite of the primarily stationary FAC structures on this quiet day, overall low ratios are obtained. This is caused by the relatively small amplitude of the signal, falling over large parts below our threshold of RMS > 2 nT/s. A quite different picture emerges from the stormy day on 4 November 2021. The example in the lower part of Figure 7 shows a northern hemisphere polar pass. Bursts of activity appear in the evening and morning sectors. Their amplitudes are an order of magnitude larger than those on the quiet day. The correlation coefficient generally stays below the threshold, $Cc = 0.75$, and also the optimal time lag is in most cases too short in comparison with the spacecraft separation. Another but minor effect is the equatorward expansion of the auroral activity during the storm to latitudes below 60° MLat and with that a movement out of our activity monitoring range. Large parts of the greatly expanded polar cap are practically free of signal. All this results in very few positive detections of stationary FAC structures during this intense storm.

All the observations presented so far indicate that there are several processes that influence the stability of small- and meso-scale FAC structures. When looking at series of satellite observations, several of the controlling parameters vary at the same time. Therefore, it is not easy to disentangle the effects.

## 5. Discussion

The main purpose of the study is to find, for FACs of small- and meso-scale sizes, their azimuthal correlation length and temporal stability. Of special interest here are the properties of the small-scale FACs, which have never been investigated in comparable details, and which are known to be associated with Alfvén wave activity.

The observations presented in the previous sections show that the correlation length of FAC structures have different dependences on spatial and temporal scales. In addition, also local time, season, latitude range and solar wind input may play a role. In this section we try to disentangle the various influences that determine the stationarity of a FAC structure. No such effort can be found in the past literature for achieving this in comparable detail.

*5.1 General characteristics of small- and meso-scale FACs*

In a first step we look at the spatial scales and temporal correlation lengths of stationary FACs. Observations presented in Figure 5 indicate that the percentage of detected stable current structures depends on both the along- and cross-track separations between the Swarm satellites. For further investigating these dependences, Figure 8 presents the occurrence ratios for all the six band passes in $d_{cross}$ versus $\Delta t$ frames. Here $d_{cross}$ denotes the cross-track separation in unit of kilometer between Swarm A and C. The individual ratio percentages have been dropped into bins of 1 km for $d_{cross}$ by 3 s for $\Delta t$. Bin averages are then shown in color. Results from both hemispheres are combined. Due to the given orbit geometry, the cross-over points are generally close to the geographic poles. Thus, only relatively small transverse separations are experienced in this high-latitude study. Within the range of $d_{cross}$ = 0 - 15 km, as shown in Figure 8, we cover almost all the available cases (only a few reach up to $d_{cross}$ = 20 km at $\Delta t$ = 6 - 9 s). The white areas in Figure 8, on the left side and in the top parts of the panels indicate constellations that have not been covered by the Swarm A and C satellites. For classifying a current event as suitable for dual-SC FAC estimates we request occurrence ratios larger than 50%, as done also in the earlier study by Zhou et al. (2024).

In general, Figure 8 confirms the impression gained from Figure 5, in particular for short period structures (small FACs); the occurrence ratios drop off rapidly away from the epoch of coplanarity. It is quite clear from Figure 8 that the patterns of occurrence ratios differ significantly between the upper and lower groups, the three shortest periods and the two longest, respectively, while the 13 - 23 s band contains something from both groups. Here we will go a little more into the details of the different characteristics observed for the two groups of FAC scale ranges.

The azimuthal correlation length of the meso-scale FACs (23-60 s or 86 - 225 km scale size) is obviously larger than the experienced satellite cross-track separations (0 - 20 km). Therefore, the along-track time difference, $\Delta t$, between the sampling by the two Swarm satellites is more decisive for the signal correlation. Also here, larger structures are stable over a longer times. Generally, the 50% demarcation lines in Figure 8 lie for the lowest two period bands beyond the covered $\Delta t$ range of 40 s. This confirms earlier studies, e.g. Gjerloev et al. (2011); Lühr et al. (2015), who reported stability periods of 60 s and more for FACs of these scale sizes. With the given limitations, the present study cannot provide much new information about the character of these meso-scale FACs at auroral latitudes. Still, it should be noticed, these results confirm well the applied preprocessing approach of the magnetic field recordings for the Swarm standard dual-SC FAC processing. For the calculations of those products the horizontal field components are low-pass filtered with a cutoff period of 20 s. This suppresses the variable small-scale signals. The along-track separation between Swarm A and C varies during normal operation between 4 and 10 s. In addition, there are the step sizes of 5 s, forming the integration quad (see Ritter et al. (2013) and Lühr et al. (2020) for more details). This means, measurements contributing to a dual-SC FAC density can be separated by up to 15 s. According to our results, the field recordings at the two spacecraft are still well correlated with each other. Figure 8 confirms that around 90% of the FAC structures with periods longer than 20 s are stable over times longer than the 15 s.

The stationarity characteristics of structures with periods of 3 to 13 s are quite different. Their cross-track size seems to be the more limiting factor than the temporal stability. For example, in

the case of the 3-7 s band we find for small satellite separations, $d_{cross} < 6$ km, ratios of about 50%
or more up to $\Delta t \sim 18$ s. (Note that the intense storm on 4 November 2021 occurs just at a time
difference of $\Delta t = 18$ s, thus masking somewhat the actual location of the 50% boundary).
Conversely, for larger cross-track separation, say $d_{cross} > 8$ km, we find nowhere in this period
band 50% ratio levels. Also in the 7-13 s bands, we find a similar tongue of elevated ratio levels
for small spacecraft separations, $d_{cross} < 6$ km. This indicates that a majority of small-scale FACs
exhibits such a narrow transverse size.
The FAC structures in the shortest period band (1-3 s) seem to belong to another, third category
of current features. Outside the $\Delta t = 2$ s separation range only very few common magnetic features
are observed by the two Swarm satellites. These small FACs with scale sizes of 4 to 11 km seem
to be stable only for a second or so. Thus, the longitudinal separation of the spacecraft has no effect
on the derived occurrence ratio. This type of very small FACs is surely worth a more detailed
investigation and will be the topic of a follow-up study.
Already Ishii et al. (1992) had reported about two types of FACs deduced from the ratio $\Delta B_{east}$
over $E_{north}$. The one type, related to longer periods in satellite recordings, was identified as stable
current structures. For such FACs it can be assumed that the field lines act as equal potential lines
between magnetosphere and ionosphere. In those cases, the $\Delta B$ over $\mu_0 E$ ratio reflects the Pedersen
conductance. These authors mentioned a short-period limit of 8 s for the stable FACs. Here we
have identified a mean transition period around 15 s as the limit for stable meso-scale FACs.
Although based on very different approaches, both studies come to fairly consistent results for the
typical scale range of stable FACs. For shorter period (smaller) current structures Ishii et al. (1992)
report a progressive increase of the E over $\Delta B$ ratio down to along-track scales of some 10s of
kilometers. They interpret that small FAC type as transient Alfvénic mode.
In a similar study Pakhotin et al. (2018) made use of Swarm constellation data for investigating
the FAC characteristics at various spatial scales. For the one event they considered they looked at
the correlation between the magnetic field recordings at Swarm A and C. For shorter FAC scales
(along-track wavelength < 75 km) they find significant differences between the readings at the two
satellites (see their Fig. 2). From this single pass observation they cannot decide whether the
missing correlation is caused by the difference in time between the two measurements ($\Delta t = 10.7$
s) or the longitudinal separation between the spacecraft (25-30 km). They guess that the
decorrelation is caused by the time delay between observations. However, our results do not
confirm their suggestion. For this class of FACs they have clearly a too large transverse separation
between measurement points. Thus, the two spacecraft are sampling two different fluxtubes.
The authors also made use of electric field estimates from Swarm A. By calculating the $\Delta B$ over
$\mu_0 E$ ratio they obtain an estimate of the apparent Pedersen conductance. As can be deduced from
their Figure 5, up to a frequency of ~0.15 Hz, constant impedance values result, for higher
frequencies the impedance increases. This obtained apparent period of ~7 s, as lower limit for
stable meso-scale FACs, is well consistent with the report from Ishii et al. (1992). The shorter-
scale FACs become more dynamic, thus partly driven by Alfvén waves, and some of the incoming
energy is reflected, which manifests itself in a decrease of the apparent Pedersen conductance. For
even shorter scales the pure Alfvén mode is approached with $E/B = V_A$, where $V_A$ is the Alfvén
velocity.

*5.2 Features of small-scale FACS*
When starting with the smallest scales (1-3 s period), we find characteristics different from those
of all the FACs of other period bands. There is practically no appreciable correlation obtained
between the recordings of Swarm A and C, regardless of their cross-track separation, except when
the along-track separation is reduced to $\Delta t = 2$ s. This strongly suggests that the life-time of these
FACs is very short, order of 1-2 s. This infers that they belong to the class of very small-scale
FACs, as described before by Neubert and Christiansen (2003); Rother et al. (2007). In practice
this means, the approach applied here is not suitable for investigating that class in more details.
Of particular interest for this study are the FACs in the period bands of 3-13 s. First, we may look
into the distribution of stationary events. Figure 9 shows in the upper two rows how the amplitude
of the magnetic variations for selected FAC structures is distributed in magnetic latitude over our
study period. The 3-7 s period band (~10-25 km scale) is used as example for this class of FACs.
Only positive event detections within the above-described range, $d_{cross} < 6$ km and $\Delta t < 18$ s, are
considered. In the following we use the term "selected" for those current structures in the above
defined scale range that passed the stability checks and "deselected" for those not passing the
checks. Separate frames display the results from up- and downleg orbital arcs and from the two
hemispheres. The blank areas reflect the lack of entries due to our spatial/temporal constrains, e.g.,
at lower latitudes and at times of large spacecraft separations. Prominent stationary current
structures appear in the northern hemisphere near 80° MLat in the morning to noon sector during
the first half of the study period. At the same time, much less activity is observed in the southern
hemisphere with smaller amplitude and at somewhat lower latitude. We attribute these
hemispheric differences primarily to the season, with local summer in the north and winter in the
south. During night-time hours events appear at much lower latitudes and significantly smaller
amplitudes. The second half of the study period is rather sparsely populated due to our constrains.
Here we have southern summer conditions. As expected, more significant FAC activity appears in
this hemisphere on upleg orbital arcs during morning and prenoon hours. Although we have winter
in the northern hemisphere, it still shows appreciable small-scale FAC amplitudes in this time
sector. The downleg arcs exhibit in the south some moderate FAC activity in the auroral region
during pre-midnight hours. In the northern hemisphere the sampling is too sparse in this time
interval for providing additional information.
In comparison to the identified stationary small FACs, the two lower rows of Figure 9 show the
amplitude distribution of the deselected current events within the above-described range, $d_{cross} <$
6 km and $\Delta t < 18$ s in the same format. An obvious difference between the two distributions is the
generally larger amplitude of the deselected events. This difference in amplitude is more
pronounced on the nightside than on the dayside. The latitude distribution of deselected events is
clearly narrower than that of the selected. This suggests a preferred stability of the smaller
amplitude FAC structures located away from the latitude of peak small-scale FAC activity.
There may, however, be a different explanation for the appearance of deselected events at certain
latitudes.  From earlier studies, e.g., Neubert and Christiansen (2003); Rother et al. (2007), it is
known that also very intense FACs with horizontal scales of about one kilometer exist at those
latitudes. These FACs appear quite randomly within bursts. As a consequence, the large current
spikes exhibit an almost white signal spectrum, up to 8 Hz, in magnetic field recordings, as shown
by Rother et al. (2007). This means, the km-scale FAC activity may also influence the signal in
our 3-7 s period band. For checking that hypothesis, we looked at the ratio between amplitudes of
the 1-3 s and 3-7 s periods signals. Here we take the intensity of our shortest period, the 1-3 s band,
as representative for the km-scale FAC activity, although we know that prominent FACs of clearly
shorter periods exist. The ratio of $RMS_{1-3s}$ over $RMS_{3-7s}$ is calculated separately for the selected
and deselected events of the 3-7 s period in the $d_{cross} < 6$ km and $\Delta t < 18$ s range. The distributions
of ratios between these event types show two clearly separated functions, see Figure 10. From the
fitted Gauss curves we obtain for the selected events a mean ratio of 0.65 ±0.22 and for the
deselected a somewhat broader distribution with a ratio of 0.93 ±0.29.
In the quest for reasons that may be in favor for large km-scale current structures we checked the
prevailing IMF and solar wind conditions during the study period. As a parameter for representing
the amount of solar wind input, we choose the merging electric field, $E_m$, as defined by Newell et
al. (2007) (they call it coupling parameter)

$$E_m = \frac{1}{3000} V_{SW}^{\frac{4}{3}} \left( \sqrt{B_y^2 + B_z^2} \right)^{\frac{2}{3}} \sin^{\frac{8}{3}} \left( \frac{\theta}{2} \right) \tag{4}$$

where $V_{SW}$ is the solar wind velocity in km/s, $B_y$ and $B_z$, both in nT, are the IMF components in
GSM coordinates, $\theta$ is the clock angle of the IMF. With these units the value of the merging
electric field will result in mV/m. Here solar wind and IMF data, averaged over 1 min and
propagated to the bow shock, are used to obtain a weighted time-integrated merging electric field
which accounts for the memory effect of the magnetosphere-ionosphere system. Details of the
approach can be found in the publication of Zhou et al. (2018).
Figure 11 shows among others the *Em* values averaged over a day for our whole study period. For
most of the time only low to moderate *Em* values are prevailing, staying commonly below 2 mV/m.
In addition, there are curves in Figure 11 reflecting the ratios of RMS$_{1-3s}$ over RMS$_{3-7s}$ separately
for the selected and deselected events. The up- and downleg orbital arcs show results from different
local time sectors. Also these values are daily averages. The data gap of the ratios, November and
December 2021, is due to the large along-track spacecraft separation, exceeding $\Delta t = 18$ s. The
ratios resulting from the selected events follows, independent of season and local time, an almost
straight line of constant level around 0.7, consistent with the distribution curve in Figure 10. The
ratio curve for deselected events is more variable but stays for all the times above the value of the
selected. A comparison between the *Em* curve with those for the ratios shows not obvious
correlation. There seems to be no direct influence of the solar wind input on the size of the ratios.
In spite of that, when looking at the actual *Em* values at the individual times of selected or
deselected events, we find systematically a slightly larger merging electric field of about 0.2 mV/m
for deselected cases.
The features of the obtained distributions are in favor of our suggestion that large km-scale FACs
can compromise the correlation of the 3-7 s period signals at the two spacecraft. As mentioned by
Rother et al. (2007), the intense km-scale FACs tend to come as solitary current spikes, thus
causing an almost white signal spectrum. The spectral leakage from these spikes into the longer-
period bands will markedly contribute to the 3-7 s period signal. Due to the short life-time of the
spikes, of order one second, it will contaminate the 3-7 s signal at the two Swarm satellites in
totally different ways, because only for the retarded signals of Swarm C peak cross-correlation
coefficient are achieved. In a way it is like adding independent series of white noise to the signals
at the two spacecraft. Thus, it is no surprise that the cross-correlation coefficient drops below the
threshold in those cases where the signal from the km-scale FAC exceeds the amplitude of the 3-
7 s by a certain factor. All this suggests that a majority of these small-scale events have falsely
been deselected by our approach.
Conversely, we may suggest that all the small-scale FACs within the given spatial and temporal
limits are viable events. Such a case is presented in Figure 12, showing the amplitude distribution
in the same format as Figure 9. When comparing the two figures, nothing has changed in the local
time and latitude distribution, just the peak values have become slightly larger (note the change in
color scale). This confirms our suggestion, the environmental conditions that increased the
amplitudes of the 3-7 s period signal, even stronger amplified the km-scale FACs. As a
consequence, the spectral leakage from this latter class caused that we just deselected the large-
amplitude small-scale FACs. When taking all that into account, we think, Figure 12 provides a
more realist amplitude distribution.

From the results obtained so far, we suggest that in the auroral zone the class of small-scale FACs in the 3-13 s period range (10-50 km scale) represents an individual class. A majority of them is detected in the spatio-temporal range, $d_{cross}$ < 6 km and $\Delta t$ < 18 s. The total number of events within the limited range amounts to around 390,000 for the 3-7s period band. In comparison, all the events outside that range are about 230,000. From those on average only 20% are selected. That results in a relatively small number of 46,000 stationary cases with larger temporal and/or spatial correlation lengths. These facts are in favor of the small spatial correlation length of this small-scale FAC type. Following the probability considerations, as outlined by Xiong and Lühr (2023) and Zhou et al. (2024), the mean correlation length of the current structure is expected to be twice as long as the spacecraft separation at the 50% occurrence ratio. In our case this means the obtained distance of 6 km infers a 12 km scale size in azimuthal direction. From these results we may conclude that this class of small-scale FACs at auroral latitudes exhibits a filamentary character with short transverse correlation lengths.

*5.3 Possible drivers for the small-scale FACs*

We may ask, which magnetospheric processes are responsible for such filamentary FAC structures. In the literature several suggestions can be found. We may start with the noon time, where the small-scale FACs appear particularly frequent. In this local time sector flux transfer events are believed to be the main source of transient and filamentary FAC structures. They manifest themselves optically as poleward moving auroral forms (e.g., Lockwood et al., 1990; Omidi and Sibeck, 2007). The related field-aligned currents are expected to cover spatial scales down below 100 km, thus fitting into the 3-13 s period range. On the duskside a viable generation process for transient filamentary FACs is the formation of Kelvin-Helmholtz plasma vortices. They are a result of strong plasma flow sheer between the magnetosheath and the magnetosphere in the range of the LLBL. According to Johnson et al. (2021) filamentary FACs of scales 50-100 km in the ionosphere are expected to connect to the vortex centers at the LLBL. For the example they present, they find scales of about 70 km. Also this fits into the range of our class of small FACs. An example for transient phenomena on the morning side are the travelling convection vortices (TCV) (e.g. Friis-Christensen et al., 1988; Lühr et al., 1998). They are caused by local pressure pulses in the solar wind causing undulations on the magnetopause that move with the solar wind from the day to the nightside. Related ionospheric effects propagate from the prenoon sector to the morning side. The magnetopause undulations are coupled by a pair of oppositely directed FACs to the ionosphere. Unfortunately, there exist so far no reports of FAC observations by satellite that could be related directly to TCV observation. Even though, filamentary FACs structures with scales of less than 100 km in the ionosphere are expected from this phenomenon. The dynamic nature of all the mentioned processes infers that they are coupled with kinetic Alfvén waves.

For these small-scale transient events we find a stability limit of 18 s. This implies a dynamic character of the current system, but the period of possible variations has to be longer than 36 s. When a reflection layer for the Alfvén waves is assumed in the magnetosphere, it has to be sufficiently far away from the Earth to accommodate derived travel time of more than 36 s.

Besides the above-described class of transient short-scale current structures occasionally stable small-scale FACs are observed exhibiting larger correlation lengths. In Figure 8 we find in some bins slightly enhanced occurrence ratios (10%-20%) for $d_{cross}$ = 12-15 km in the period band of 3-7 s. Those events seem to depend only little on the along-track separation between the satellites. A possible explanation for those cases is that they are part of the large-scale currents generation at the LLBL, as described by Johnson and Wings (2015), but caused very local density variations in the LLBL that can generate small-scale FAC structures, in the ionosphere which are stable over a longer time. Another group of small-scale FACs with larger correlation length is found near the

lower border, 60° MLat, of our considered latitude range. This second population of stable small-scale FACs is obviously related to subauroral phenomena, distinctly different from the ones at high latitudes. It would require additional investigations to characterize the features of this second population.

*5.4 The roles of correlation lag-time for the deselection*

There is another question that we may address here. What kind of current variation causes the deselection of an event? The criteria for a stationary current structure are, an amplitude RMS > 2 nT/s and a peak cross-correlation coefficient above 0.75 at a time lag agreeing with the along-track separation of the satellites within ±1.5 s. By checking, which of the criteria causes the deselection, one may obtain information about the kind of variation. A general finding is that a low $Cc$ is involved in the large majority of deselected events. In the cases of a very low $Cc$, it cannot be expected that the right T-lag is found. Therefore, a dropout of both criteria can also frequently be found. From this observation we infer that the large percentage of deselected small-scale events (up to period of 15 s) (see Fig. 8) is caused by a temporal or spatial change of the current signature between the visits by the two satellites.

In the case of meso-scale FACs high occurrence ratios, partly close to 100% (see Fig. 8), are observed. For larger along-track separations the percentage drops somewhat. In order to find an explanation for that behavior we looked at the percentage of deselected events based solely on the T-lag criterion. Figure 13 shows the distribution of that ratio in frames of $d_{cross}$ over $\Delta t$, separately for the six period bands. The color scale represents the percentage of deselections by T-lag relative to the total number of deselected events. As mentioned before, the T-lag criterion plays hardly any role for the small-scale FACs. For shorter periods, up to ~15 s (<60 km scale) about 90% of the event deselection is based on poor correlation between to satellite recordings.

We find a quite different picture for the meso-scale FACs (>75 km), for them time shifts are more important for deselection. The white gaps in the longest periods represent bins that contain no deselected events. This give a general impression about the low count numbers of deselections on which these ratios are based. The fairly high percentage causes by T-lag, varying for the longest periods, around 80%, indicates that a motion of the current system along the orbit track is the main cause here. This means, the FAC pattern does not vary in time, but the system is moving equator- or pole-ward. Such motions are well-know from auroral observations. The ratios for deselections by T-lag become progressively smaller for shorter periods, but the absolute number and distribution of these deselected events is quite similar over the period range 23 - 60 s. That means, the latitudinal propagation of the current system is similar for the meso-scale FAC structures of 85 to 220 km sizes. However, the change in dominant color from period to period indicates a clear reduction of stability, also for meso-scales, in space and time increasing towards smaller sizes.

## 6. Summary and Conclusions

In this study we made use of magnetic field data from the closely spaced Swarm A and C spacecraft during the counter rotating orbit phase. This special constellation enabled us to investigate the spatial and temporal correlation lengths of small-scale and meso-scale FAC structures at auroral latitudes. A number of detailed characteristics, in particular of small-scale FACs (10 - 50 km scale size) have been verified for the first time. We identified three classes of FAC types, populating different scale ranges, with markedly different characteristics. Below the major results are summarized. We start with the stable features of meso-scale FACs (75 - 220 km scale). The main findings of the study are:

1. In general, the stability features of meso-scale FACs, reported in earlier studies, have been confirmed. For example, the actual transverse (azimuthal) correlation length is larger than our maximum spacecraft separation of 20 km. Therefore, no upper limit for that value can be provided by this study. Over the main part of the study period detection ratios close to 100% (see Fig. 5) have been derived for this class of FACs. Slightly reduced ratios were obtained in cases of long along-track spacecraft separations (up to 41 s). A closer inspection of deselected events revealed a latitudinal motion of the FAC system as main cause for the decorrelation, not a change in current structure. All our findings are consistent with the reports of Gjerloev et al. (2011) and Lühr et al. (2015) stating that meso-scale FACs are stable for 60 s or longer.

2. More interesting and new results have been obtained for the class of small-scale auroral FACs with along-track scale sizes of 10 to 50 km (3-13 s period band). For most of them an azimuthal correlation length of only 6 km is found at 50% detection ratio level. From statistical considerations it follows that a 50% probability is achieved in a random sampling when the actual correlation length is 12 km (two-times 6 km). Such a limited transverse scale size of this class of small FACs has never been reported before. For the characteristic time period of their temporal stability a typical value of 18 s has been obtained. This suggests a dynamic nature of these FAC systems probably associated with kinetic Alfvén wave exhibiting periods longer than 36 s.

3. Peak amplitudes of these small FACs are found in the noon to prenoon sector at 75°- 80° MLat. The dayside activity is significantly stronger in the summer than in the winter hemisphere. On the nightside smaller amplitudes prevail peaking at latitudes around 70° MLat. On average, more intense small-scale FACs are observed in the northern hemisphere than in the southern.

4. A third class of very small FACs (1-3s, 4-10 km) has been identified. For this class hardly any correlation between the recordings of the two Swarm satellites is found, regardless of their cross-track separation distance. We assume that these FACs exhibit a very short life-time of order 1 second. Therefore, appreciable correlation (>50% ratio) is only achieved when the along-track time difference between the spacecraft is reduced to 2 s. The study approach used here is therefore not suitable for deriving more details about that class of FACs. From earlier publications, e.g., Neubert and Christiansen (2003); Rother et al. (2007), it is known that these km-scale FACs can reach large amplitudes. A detailed investigation of these very small FACs will be the topic of a follow-up study.

5. An investigation of the deselected small-scale (10-50 km) current structures reveals that those with large amplitudes are commonly accompanied by FACs of some kilometer scales. This observation suggests that the correlation between the magnetic signals at the two Swarm satellites is compromised by spectral leakage from the large-amplitude narrow current spikes into the considered period range of 3-7 s. Thus, a part of our small-scale FAC events is probably falsely deselected due to the limitations of our approach. We could show that our small-scale FACs (3-7 s period) are commonly deselected when their amplitudes are smaller than those of the km-scale FACs. We thus preferred to accept all small-scale FACs in the $d_{cross}$ < 6 km and $\Delta t$ < 18 s limits as stationary.

6. There exists a minor population of small-scale FACs that is stable over longer time, $\Delta t$ < 40 s. Also, its azimuthal correlation width seems to be longer than 15 km. The preferred appearance is at the equatorial end of our latitude range, near 60° MLat. No detailed investigations of these subauroral FACs have been performed, but they seem to be worth considering in a dedicated study.

From the results listed here we can conclude, the large-scale FACs are stable for more than 40 s. Reason for their occasional deselection is mainly the latitudinal motion of the current system, while their shapes remain constant. Meso-scale FACs (50-150 km scale size) also show the effects of latitudinal motion, but in addition, the shape starts to vary within 40 s. The smaller structures exhibit more variation. The small-scale auroral FACs (10-50 km scale) exhibit quite different

characteristics. They behave more like waves. Stationary events are only found in a limited range of spacecraft separations both in time and spatial differences. Their stability is shorter than the Alfvén transition time from the magnetosphere to the ionosphere. The very small-scale FACs with order of kilometer scale lengths, falling below the resolution of this study. Even though, they influence the signal of the longer period FACs. For that reason, they will be the topic of a dedicated follow-up study.

*Editorial note:* For obtaining more information about the announced follow-up study, a preprint of the manuscript, "Properties of large-amplitude kilometer-scale field-aligned currents at auroral latitudes, as derived from Swarm satellites" by Zhou and Lühr is available at:
https://editor.copernicus.org/EGUsphere/ms_records/egusphere-2025-1961

*Author contributions*. HL has outlined the structure of the study, YZ carried out the data processing. Both authors participated in writing the manuscript.

*Competing interests.* The authors declare that they have no conflict of interest.

*Code and data availability.* The authors thank the European Space Agency for openly providing the Swarm data. The data products used in this study are Level-1b MAGx_LR with version number 0602, which are available at the European Space Agency website: https://earth.esa.int/web/guest/swarm/data-access. There are no specific codes used for this study.

*Acknowledgments.* The work of YZ was supported by the National Nature Science Foundation of China (42174186).

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

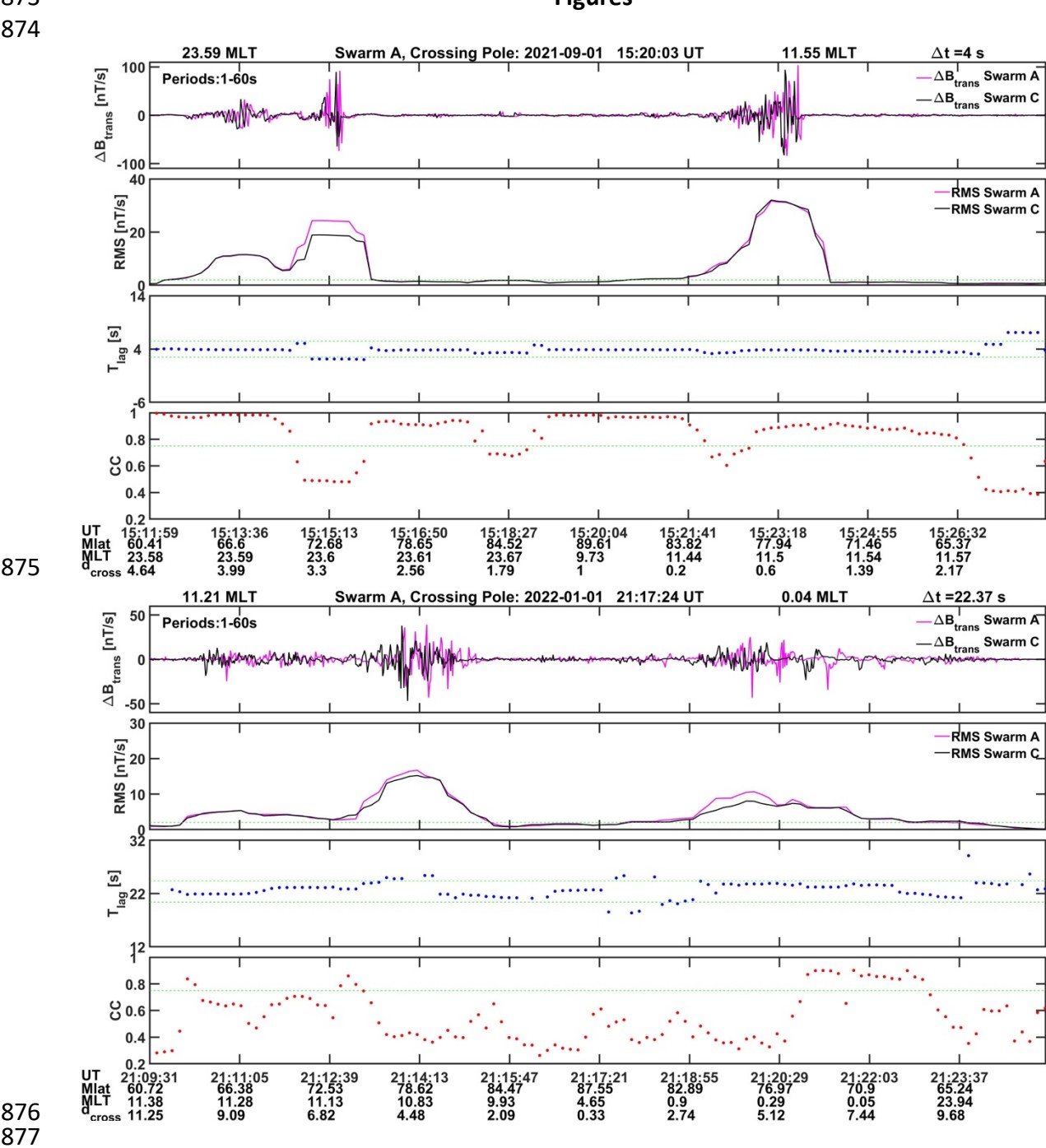




**Figure 1.** Examples of magnetic variations in the transverse, $\Delta B_{trans}$, component within the period
range of 1-60 s. The top panels of the two frames show the recordings of Swarm A and C along
their orbits, crossing the polar region of the northern hemisphere. The second panel reflects the
RMS value of the signal amplitude. The third panel contains the peak cross-correlation coefficient,
$C$c. Significant correlations between Swarm A and C should have values, $C$c > 0.75. The dashed
green lines mark the thresholds of the three detection criteria. The bottom panel shows the lag
time, T-lag, between the signals at optimal correlation. T-lag values between the two green dashed
lines indicate delays equal to the time separation between the spacecraft. The top frame is from a
time with small along-track separation, $\Delta t = 4$ s and the bottom frame from a larger separation, $\Delta t$
$= 22.3$ s

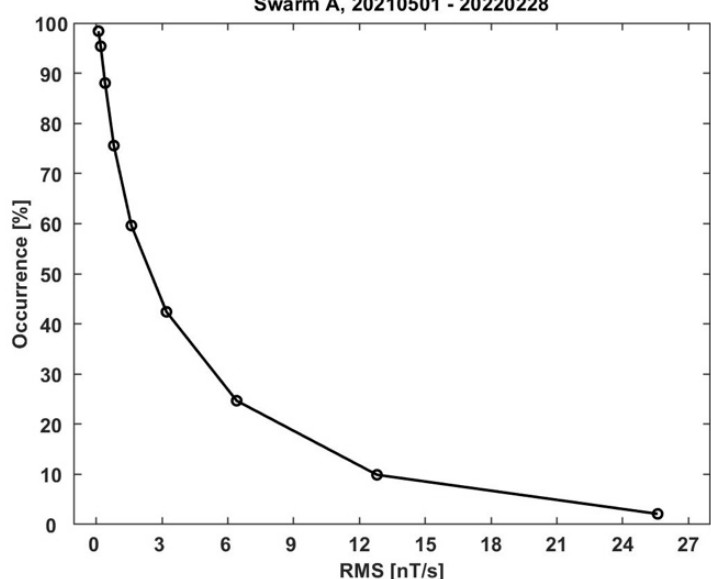

**Figure 2.** Occurrence distribution of transverse magnetic signature variations in the auroral region
with amplitudes above a certain RMS value. Here, signals in the period range 1- 60 s (4 - 220 km
along-track scale size) have been considered. For our FAC study amplitudes, RMS > 2 nT/s, are
required. With that, about 50% of the signals are taken into account.

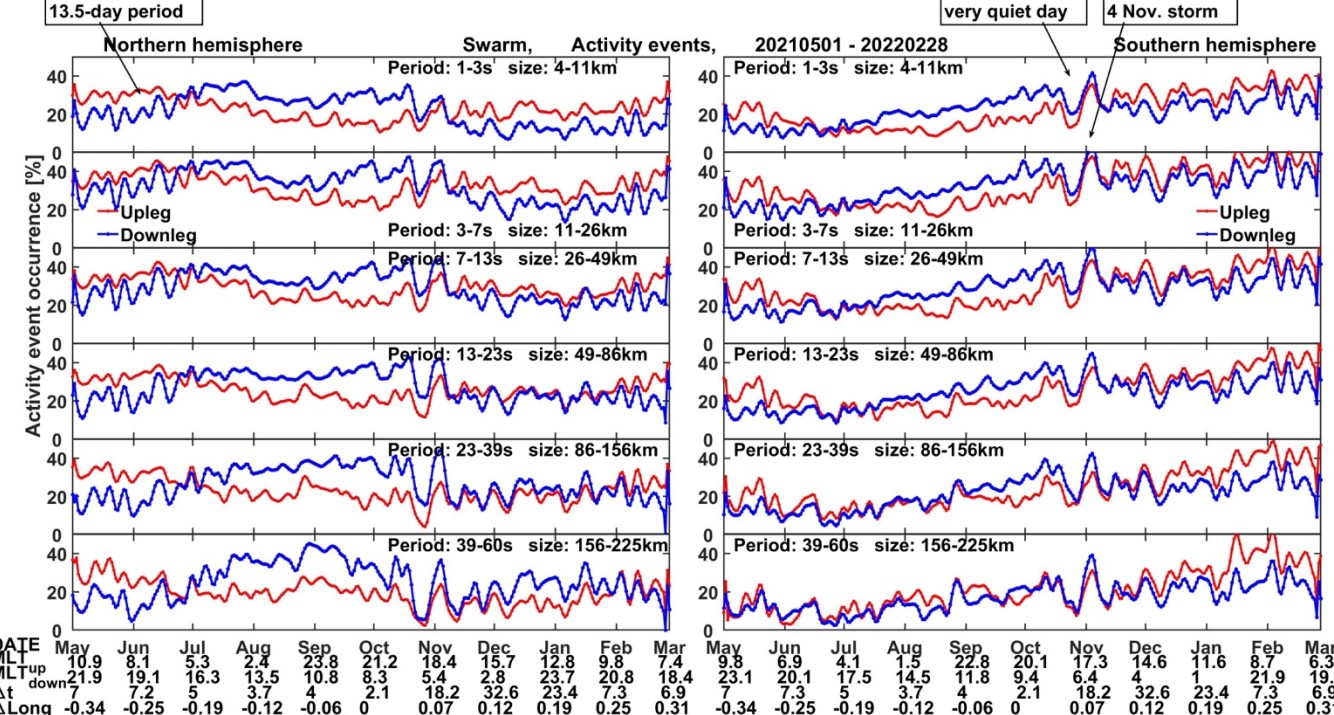

**Figure 3.** Temporal variation of all wave structures in the $\Delta B_{trans}$ component with amplitudes of RMS > 2 nT/s over the full study period, separately for the 6 period ranges and the two hemispheres. Results from the orbital upleg arcs at high latitudes (>60° MLat) are shown as red curves, those from downleg arcs are in blue. Across the bottom, besides the date, the local times at 70° MLat of the orbital arcs are listed, as well as the along-track time difference, Δt in seconds, between the spacecraft and the varying longitudinal separation at the equator, ΔLong, in degree.


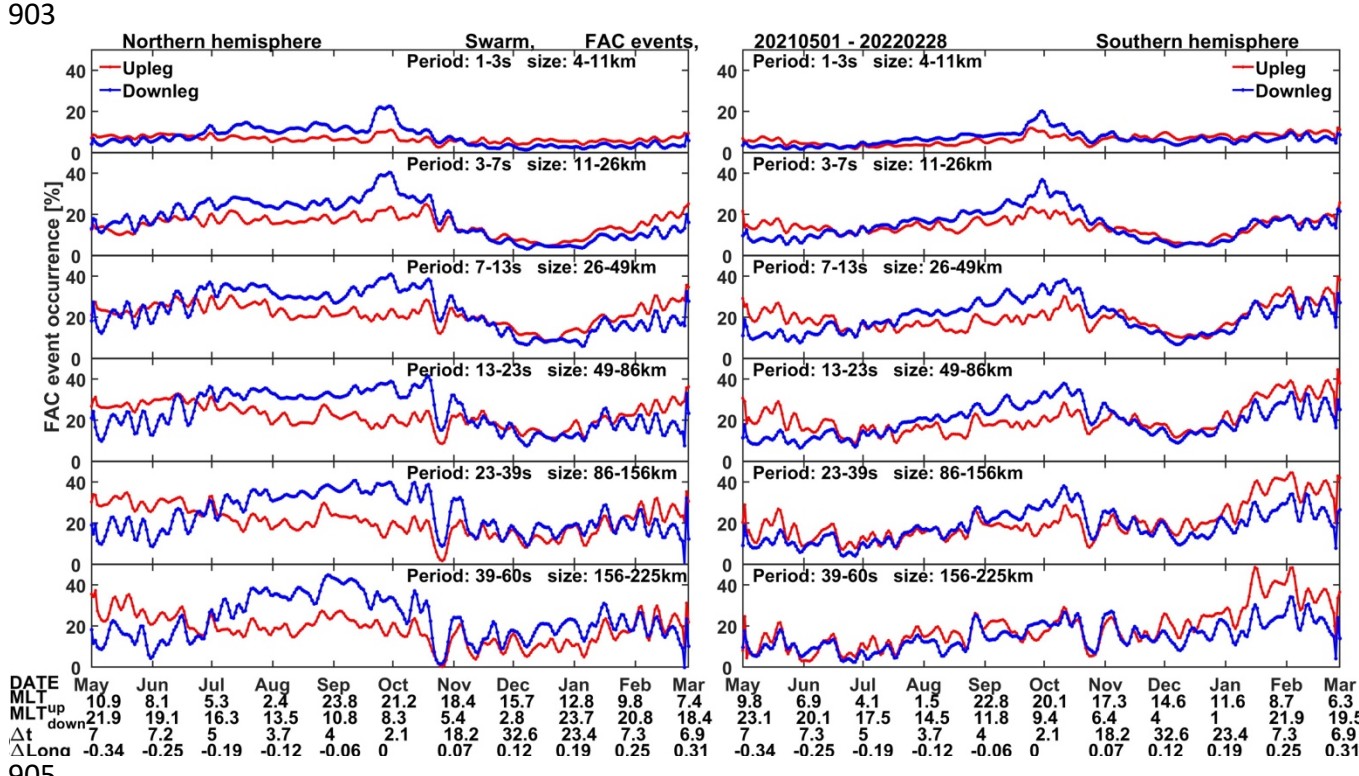


**Figure 4.** The same format as Figure 3, but for the occurrence frequency of positively detected
static FAC structures.

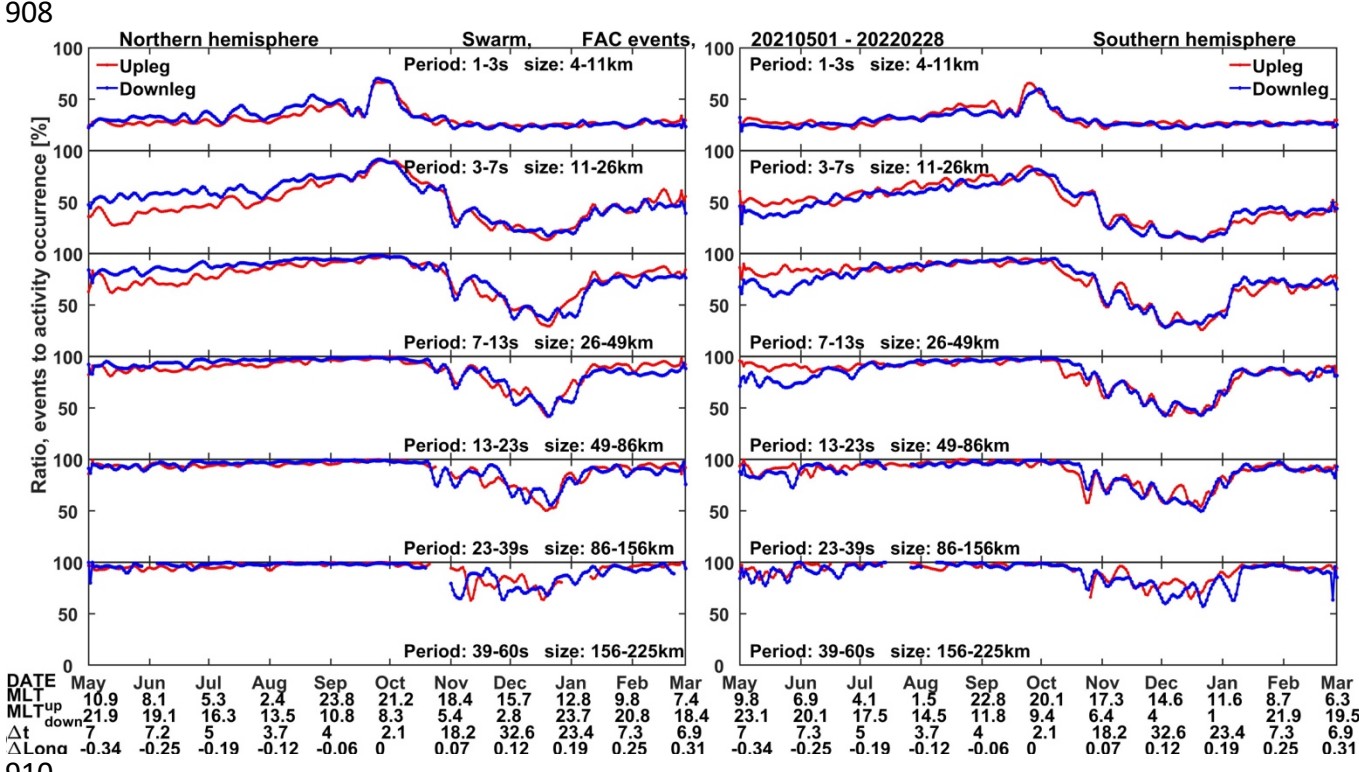

**Figure 5.** The same format as Figure 3, but for the ratio of detected static FAC structures (Fig. 4) divided by all the number of wave structures presented in Figure 3.

913

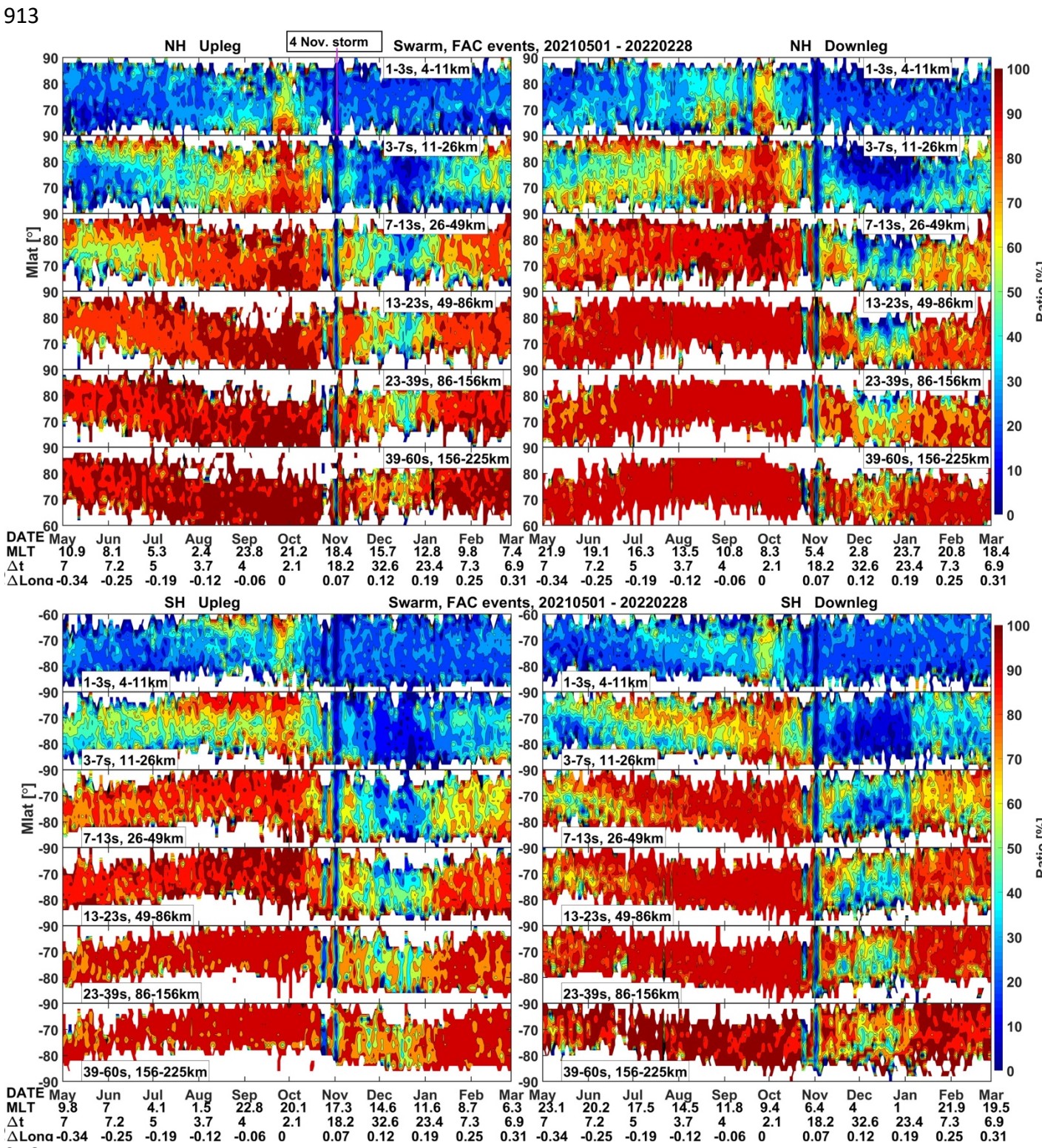

916

**Figure 6.** Latitude distribution of the ratio between positively detected static FAC structures and all wavy signals above the threshold of RMS > 2 nT/s, separately for all the six period bands, up- and downleg orbit arcs, and the two hemispheres. The labels along the horizontal axis are as described for Figure 3.


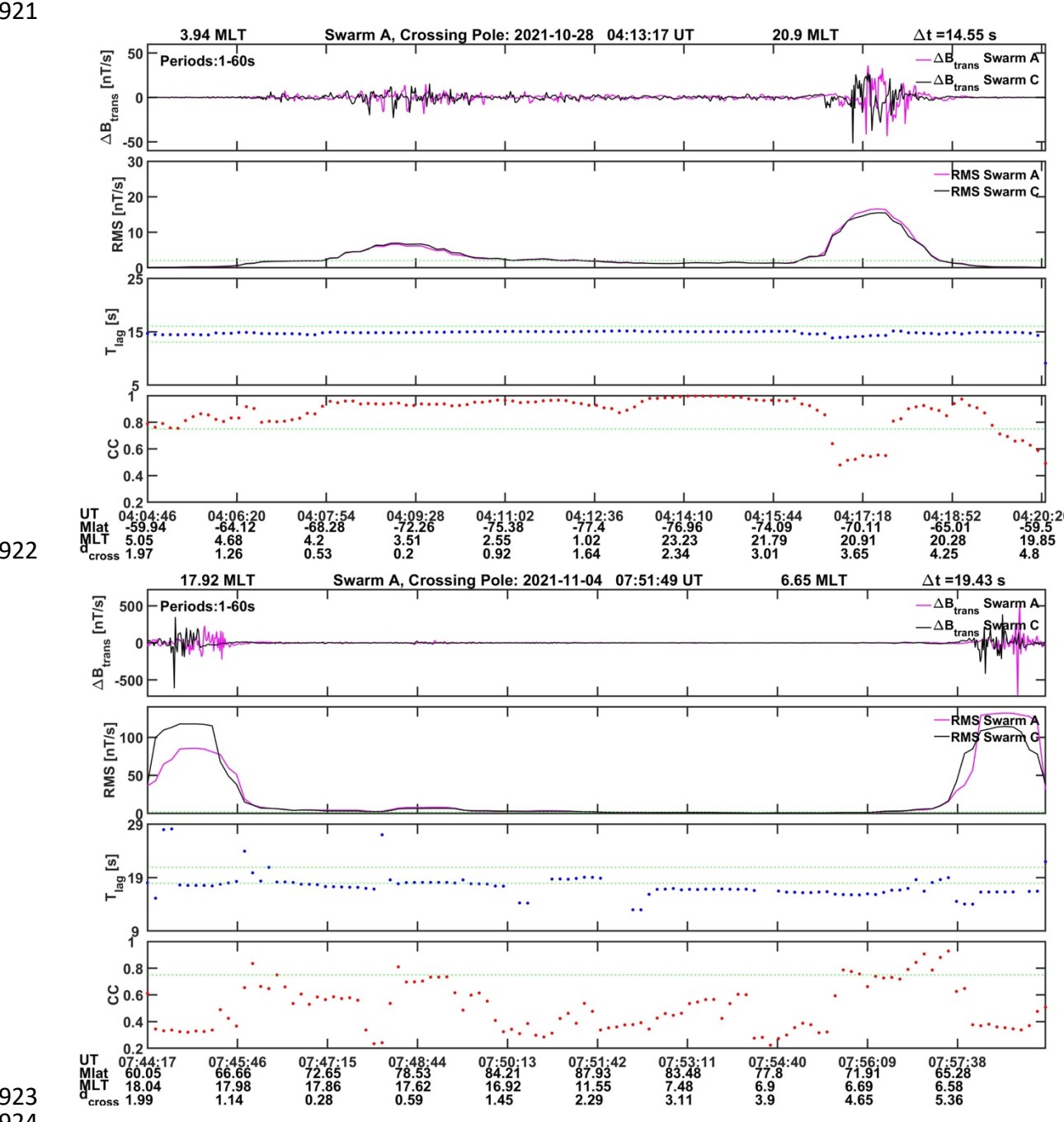



**Figure 7.** Magnetic variations in the transverse, $\Delta B_{trans}$, components within the period range of 1-60 s in the same format as Figure 1. The top frame is from a very quiet day, while the bottom frame presents variations of the storm period on 4 November 2021. Peak amplitudes differ by a factor of 10 between the two days.


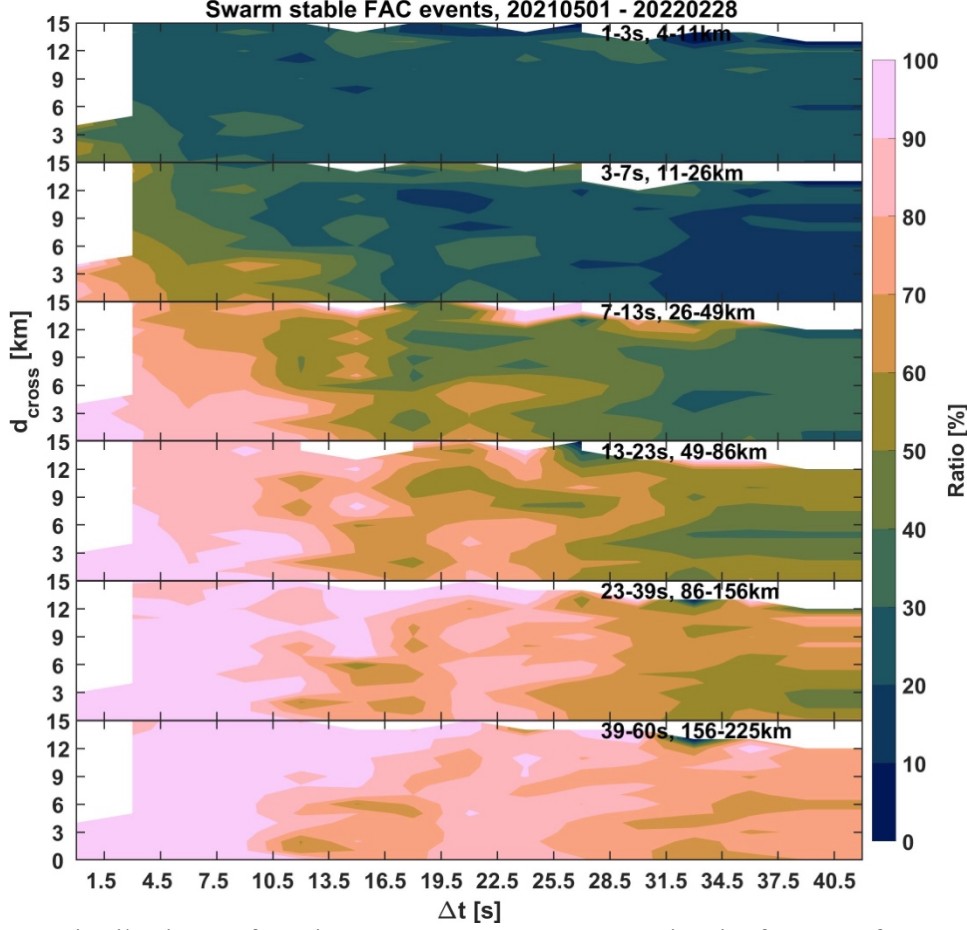

**Figure 8.** Distributions of stationary FAC occurrence ratios in frames of cross-track distance,
$d_{cross}$, versus along-track time separation, $\Delta t$, separately for the six period bands. White areas
represent parameter constellations that are not covered by the Swarm constellation.

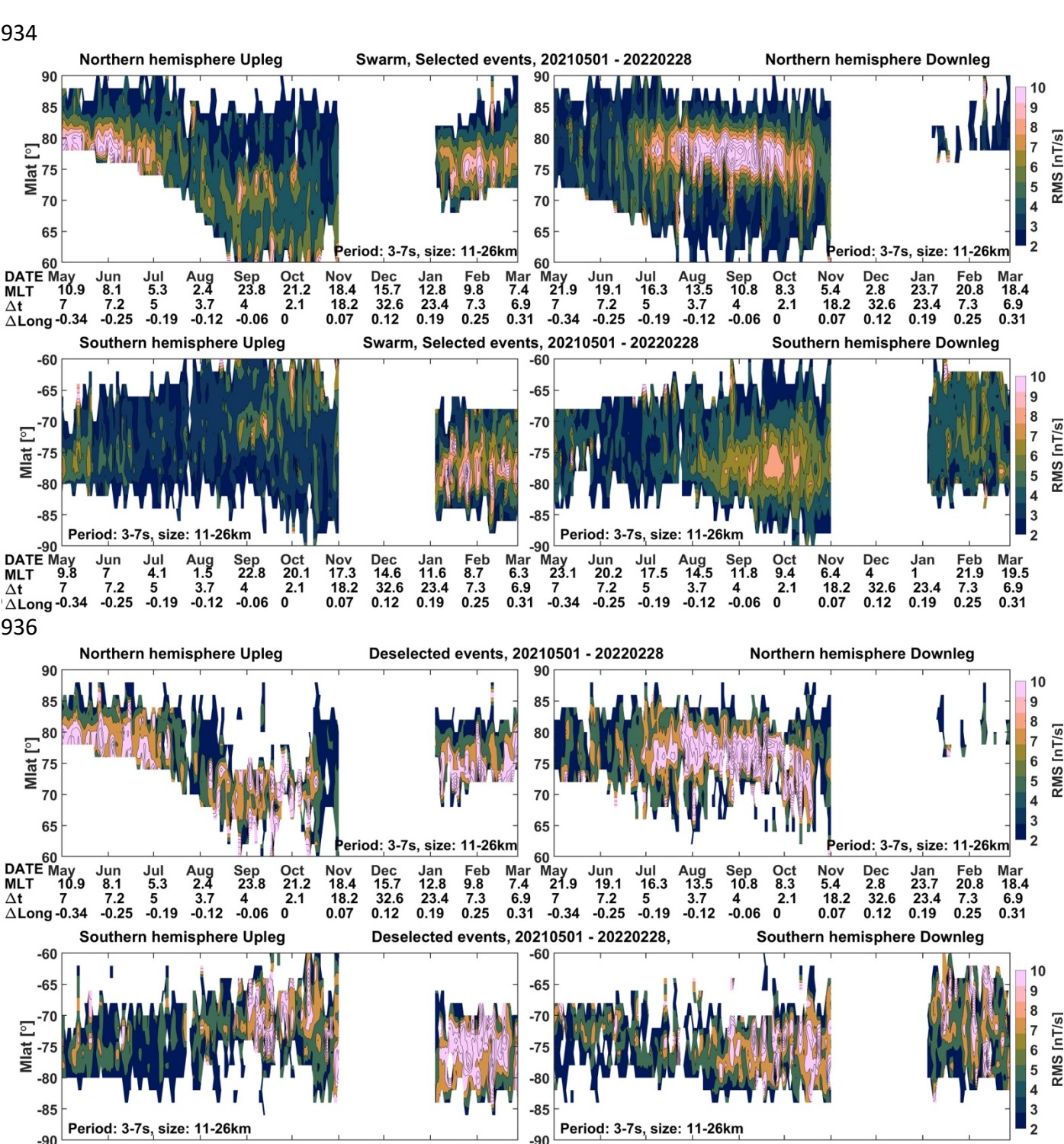


**Figure 9.** Amplitude distribution of small-scale magnetic variations over magnetic latitude. The format is the same as that of Figure 6, but only events from the 3-7 s period band and within the $d_{cross} < 6$ km and $\Delta t < 18$ s range are considered. The top two rows present the distribution of selected stationary current structures and the bottom two row the deselected events. White areas mark ranges without entries.

944

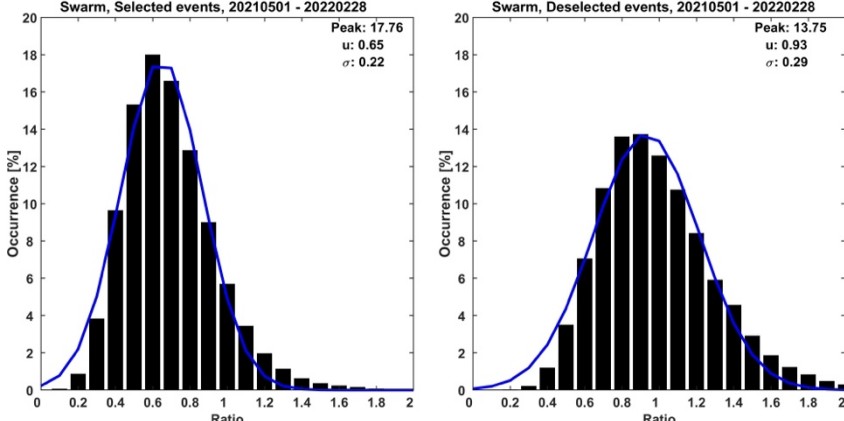

**Figure 10.** Distribution of the amplitude ratio $RMS_{1-3s}$ /$RMS_{3-7s}$ separately for the selected (left) and deselected (right) events in the $d_{cross} < 6$ km and $\Delta t < 18$ s range. The fitted Gauss curves reveal two clearly separated functions. For the selected events a mean ratio of $0.65 \pm 0.22$ is obtained and for the deselected a broader distribution with a peak at $0.93 \pm 0.29$.

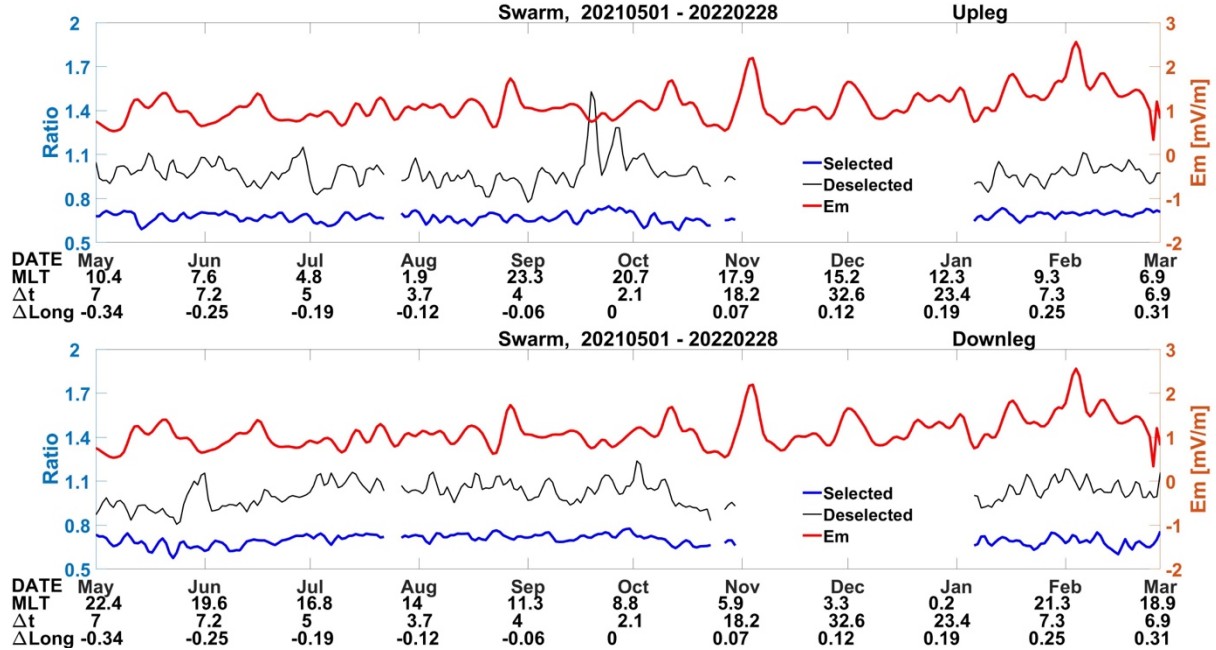

**Figure 11.** Temporal variation of the amplitude ratio $RMS_{1-3s}/RMS_{3-7s}$ separately for the selected (blue) and deselected (black) events in the $d_{cross} < 6$ km and $\Delta t < 18$ s range. For comparison, daily averages of the merging electric field, Em, are added as red curve. Both results from upleg and downleg orbital arcs are shown.


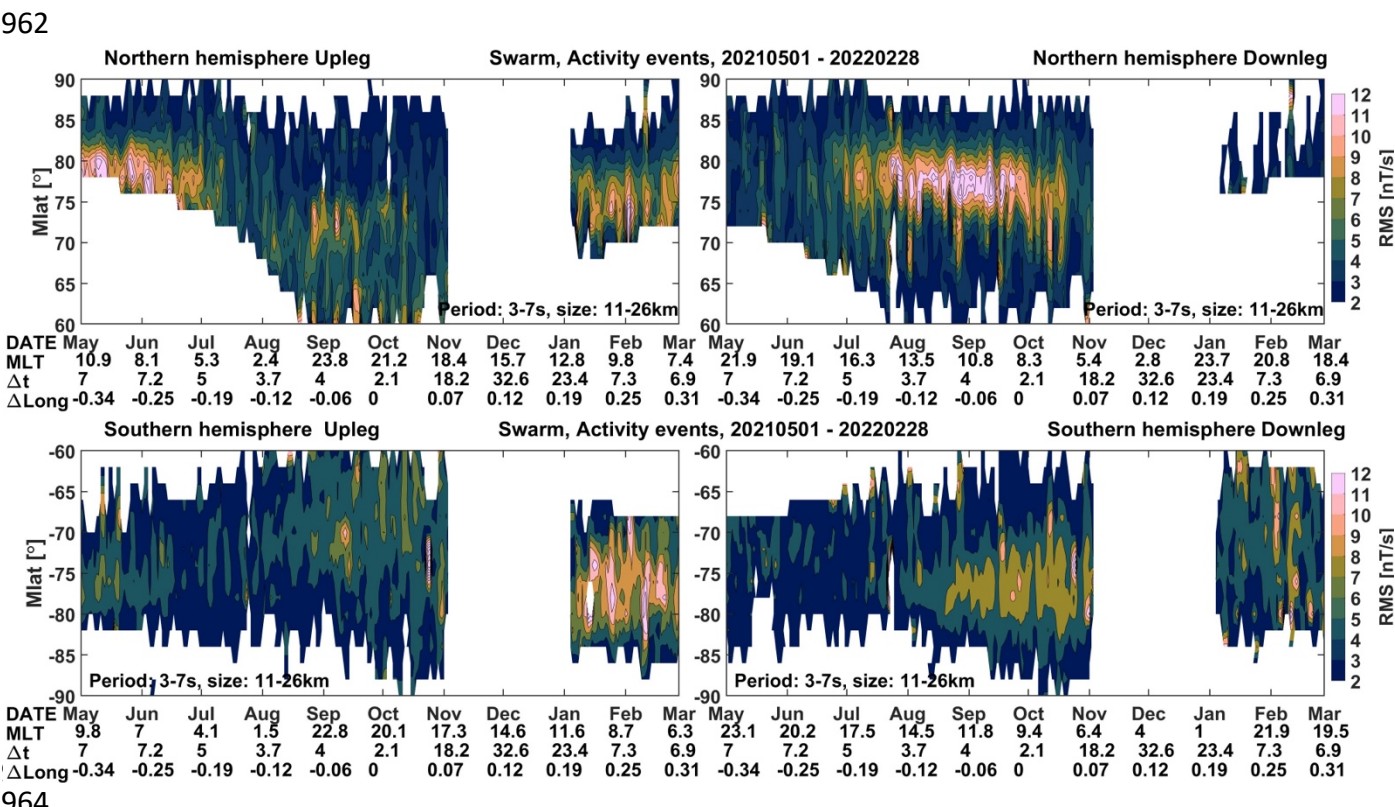


**Figure 12.** The same as Figure 9, but the amplitudes of all the events within the range of $d_{cross}$ <
6 km and $\Delta t$ < 18 s are shown.

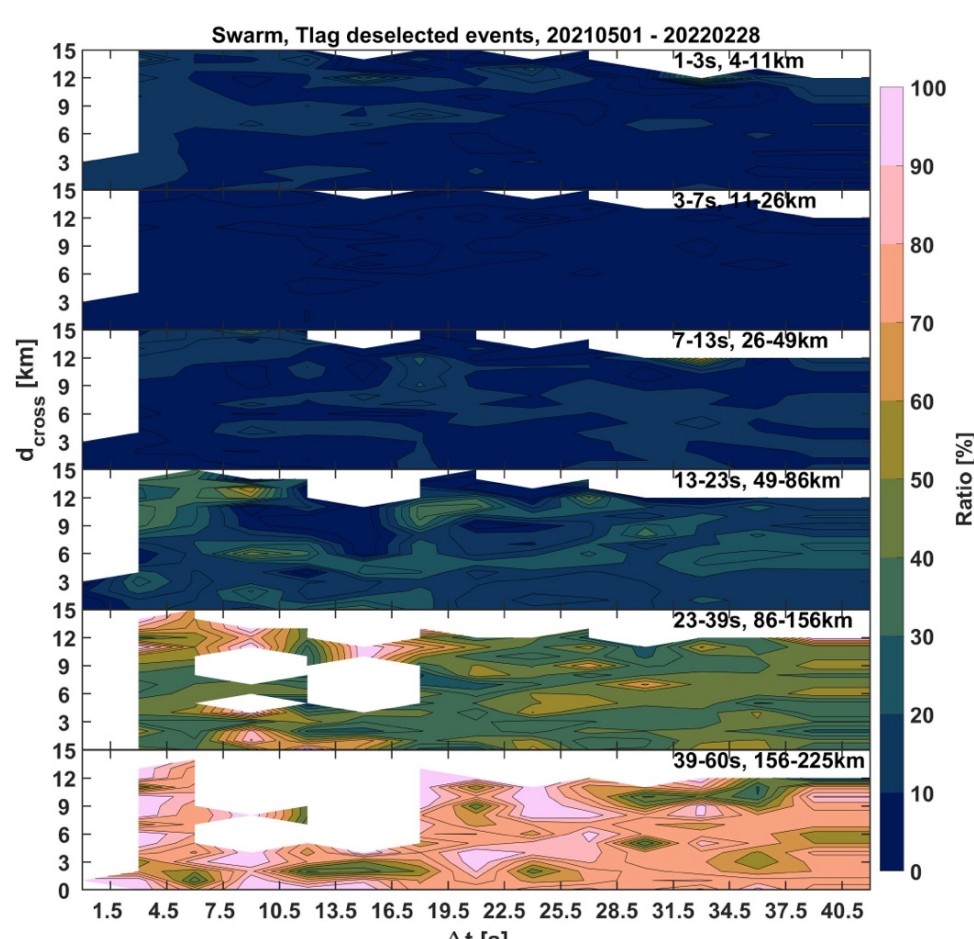

**Figure 13.** The same format as Figure 8, but for the ratio of deselected events that is solely based
on the T-lag criterium. White patches represent bins without any deselection of this kind. The T-
lag criterium is of significance only for the meso- and large-scale FACs. Low cross-correlation
coefficient values clearly dominate the deselection of small-scale current structures.