# Peer review of "Small- and meso-scale field-aligned auroral current structures, their spatial and temporal"

_Annales Geophysicae, 2024_

## Author Comment (AC1)

**Responses to the Reviewer #1 comments on the manuscript [angeo-2024-28]**

"Small- and meso-scale field-aligned auroral current structures, their spatial and temporal characteristics deduced by Swarm constellation"

by Hermann Lühr and Yun-Liang Zhou

We are grateful to the reviewer for their thorough review of our manuscript. Their constructive comments has significantly helped us to improve the study. We have responded to all the comments and made changes wherever we regarded them as appropriate. Below, please find our point−by−point replies. For the convenience of the reviewers, we have first repeated the reviewer's comment and then give our answer in blue text.

I find that the results are largely compelling, however I have two major concerns: (1) the study uses a 60 min window to capture the variability of the magnetic field which has been pre-filtered in various pass bands from 1-3s to 39-60s - this fixed window length may introduce an inadvertent bias to the results; (2) the short-period variations that are not well correlated between the two spacecraft passes are discussed as unstable field-aligned currents, but as this is a time-varying magnetic field measurement rather than a direct measurement of the current, it is unclear that this is the only possible interpretation of these results. Given that point (1) above requires some study into the methodology, I recommend that this manuscript be considered after major revisions.

We are pleased by the generally positive rating of our study. Some of the major comments are based on an obvious misunderstanding and others are considered as justified. We are responding to all of them and suggest revisions of the manuscript in most the cases. Overall, the manuscript has been subjected to a major revision. Besides many clarifications it has become more focused on the temporal and spatial properties of the small-scale (10-50 km scale size) FAC. These have never before been studied in comparable details.

Major Comments:

1) On the methodology

As noted above, I have a concern with regards to the filtering and windowing of the data in the analysis performed. The manuscript details that various time-scales of magnetic field fluctuations are considered in order to examine different scale sizes of potential field-aligned currents (derived from the spacecraft motion in that time). However, the window used to calculate the cross-correlation between the two spacecraft and RMS values of the field remains fixed. As a result, for the smallest scales, there are between ~20 and 60 possible perturbations within the 60s window whereas for the largest scales it is 1-1.5 perturbations. If a some of the small-scale perturbations vary, then the cross-correlation will drop, whereas the whole perturbation has to vary for the largest scales. This, I feel, naturally results in fewer good correlations at the smaller scales. I suggest that the authors explore the sensitivity of their results to varying the length of the window over which they do the cross-correlation. For example, are does the success ratio for the 1-3 s scales increase for a 15 s cross-correlation window. The authors may also need to consider whether the correlation coefficient threshold needs to be increased with the smaller window size such that the P-value is consistent.

We are sorry to note that the reviewer obviously misinterpreted our approach to derive the correlation between magnetic field recordings of the Swarm A and C spacecraft. For example, it is not true that a fixed time interval (e.g. 60 s) has been used for the analysis of the various filter bands. To improve the situation, not just referring to earlier papers, we have added Table 1 at the end of Section 3 listing for the various filter bands the scale sizes, the interval lengths and step sizes.

With regards to the correlation coefficient used, I note that a correlation coefficient of 0.75 means that only 56% of the variability observed by one spacecraft is observed by the other. This feels quite low to say that the observed current systems are stable. There is also no consideration of changes of amplitudes of current that might result in relatively high cross correlations but differences in RMS values.

We agree with the reviewer that a correlation coefficient of 0.75 means that only little more than half of the signals at the two spacecraft agree with each other. However, in a statistical study like this, thresholds have to be introduced. It can be found in many studies that signals are considered as correlated, as long as their Cc is above 0.71. Following those arguments, we prefer to keep our threshold Cc > 0.75. Concerning the amplitude variations between the recordings at the two locations, we deliberately do not take them into account. It is expected that the FAC intensity varies with longitude (along the current sheet). As long as the same FAC structure is encountered by the two satellites the Swarm dual-spacecraft approach (Ritter et al., 2013) will properly report the mean FAC density that is crossing the integration area. It is a secondary objective of the study to find out, which FAC types are suitable for the dual-spacecraft approach.

(2) Discussion of results

This study examines whether or not magnetic perturbations seen by one of the Swarm spacecraft were also seen by a second spacecraft passing through at a slightly later time and at some azimuthal separation. If this is the case, then it is reasonable to assert that these perturbations arise from the quasi-stationary field-aligned current system. However, when the conditions set are not met it does not necessarily follow that it is an unstable current system. I feel it is better to described the observations as temporally or spatially varying magnetic field perturbations. Previous studies using data from Swarm and other spacecraft (e.g. the cited Ishii+ 1992 study and the Pakhotin+ 2018 doi: 10.1002/2017JA024713 study) have shown that large amplitude small-scale magnetic field perturbations may be associated with Alfven wave activity.

Here the reviewer brings up a valid point. The main result of this study is the temporal and spatial correlation length of FAC structure of various horizontal scales. This point has been made clearer in the beginning of the Discussion section. Overall, we have tried to avoid the term "unstable".
" The main purpose of the study is to find for FACs of small- and meso-scale sizes their azimuthal correlation length and temporal stability. Of special interest here are the properties of the small-scale FACs, which have never been investigated in comparable details, and which are known to be associated with Alfvén wave activity."

Many thanks for mentioning the study of Pakhotin et al. (2018) which had escaped our attention. It is now mentioned in the Introduction
" Similarly, Pakhotin et al. (2018) made use of Swarm satellite constellation data and investigated FAC characteristics by a comparison between electric field and magnetic field

data for one auroral region crossing. They reported for their case a change of current characteristics around a period of 5 s (~40 km wavelength) from quasi-static to dynamic, which is well in line with the results of Ishii et al. (1992)."

and more details are considered in the Discussion. But those authors cannot distinguish between temporal and spatial stability scales, thus have to rely on unproven assumptions.

"In a similar study Pakhotin et al. (2018) made use of Swarm constellation data for investigating the FAC characteristics at various spatial scales. For the one event they considered they looked at the correlation between the magnetic field recordings at Swarm A and C. For shorter FAC scales (< 40 km) they find significant differences between the readings at the two satellites (see their Fig. 2). From this single pass observation they cannot decide whether the missing correlation is caused by the difference in time between the two measurements ($\Delta t = 10.7$ s) or the longitudinal separation between the spacecraft (25-30 km). They guess that the decorrelation is caused by the time delay between observations. However, our results do not confirm their suggestion. For this class of FACs they have clearly a too large transverse separation between measurement points. Thus, the two spacecraft are sampling two different fluxtubes. The authors also made use of electric field estimates from Swarm A. By calculating the $\Delta B$ over $\mu_0 E$ ratio they obtain an estimate of the apparent Pedersen conductance. As can be deduced from their Figure 5, up to a frequency of ~0.15 Hz, constant impedance values result, for higher frequencies the impedance increases. This obtained apparent period of ~7 s, as limit for stable large-scale FACs, is well consistent with the report from Ishii et al. (1992). The shorter-scale FACs become more dynamic, thus partly driven by Alfvén waves, and some of the incoming energy is reflected."

Minor comments:

Line 44: "at the ionosphere play**s** a role"

Corrected

Line 76: please provide a reference for the Swarm L2 data product

The references to (Ritter et al, 2013; Lühr et al., 2020) have been added.

Line 77: Swarm A and C do not fly side-by-side, as is a key point of the manuscript. I believe that for the L2 product, one of the datasets is lagged so that it is treated as if they are side by side.

We are now more specific in the manuscript: "... the Swarm A and Swarm C spacecraft flying almost side-by-side with only a small along-track separation of around 7 s."

Line 86: As noted above, I was under the impression that the dual spacecraft product is a 2D curlometer, not the mean of two single spacecraft FAC estimates as implied by this line.

The reviewer is right, the dual-SC FAC estimate is derived from a kind of curlometer. We have improved the wording in the manuscript to avoid the misunderstanding.
" One of the standard Swarm Level-2 data products is the FAC density estimate (Ritter et al, 2013; Lühr et al., 2020) derived from the magnetic field measurements of the Swarm A and

Swarm C spacecraft flying almost side-by-side with only a small along-track separation of around 7 s."

Line 90: "the range dual-spacecraft FAC estimates **are valid**" ?

Corrected

Line 116: Swarm A and C are not side-by-side by are lagged by a few seconds

Thank you for the suggestion, considered.

Line 122-128: The Zhou et al figure should be referenced here. In fact, I think the Zhou et al figure (or similar) should be included in the manuscript as it is crucial to the study.

We agree and have made a reference to Fig. 1 in Zhou et al. (2024). However, we do not want to reproduce that figure here because there are already a large number of figures in this manuscript.

Line 178: " For these **example passes, we use a 60 s sliding window**...". When I first read this I was confused as I thought the whole interval shown was 60 s.

The wording has been improved. Now it is clearly stated that the 60 s are a sliding window. " For this and the example-pass in the lower frame cross-correlations have been applied to recordings of the two satellites. Here we consider a sliding window of 60 s (corresponding to a distance of 450 km) of $\Delta B_{trans}$ from Swarm A and C for deriving the peak correlation coefficients, $Cc$."

Line 238: "look at the **variable** magnetic field signal". Given that the introduction discusses the separation of waves and FACs, the current wording was confusing

Thank you for the suggestion.

Line 246-256: It would be helpful to mark some of the intervals of interest on the figure.

Labels with arrows have been added to Figs. 3, 5, 6 pointing at events mentioned in the text.

Line 287: "practically all magnetic **fluctuations above 20 s** can be..."

Thank you for the suggestion.

Line 395: give the zonal length in km as well as seconds

Corresponding scale size added.

Line 397: I don't understand what is meant by "the time between samples is more decisive for the occurrence ratio"

The sentence has been improved.
"The azimuthal correlation length of the meso-scale FACs (23-60 s or 86 - 225 km scale size) is obviously larger than the experienced satellite cross-track separations (0 - 20 km)."

Line 414: My reading of the figure is that d_cross < 3 km is where the ratio exceeds 50%. Please confirm

We agree that there appears a valley in occurrence ratio at d_cross ~ 3 km for the 3-7 s period band. This slight dip below 50% is partly caused by the smoothing function. The proper and steep decline of ratio is observed beyond d_cross > 6 km. For these reasons we prefer to keep the 6 km 50% boundary.

Line 426-437: Consider also the Pakhotin et al (2018) study that examined Alfven waves using Swarm

Thank you for this advice. The study by Pakhotin et al. (2018) contains a lot of interesting features of our small-scale FACs. By comparing E- and B-field variations they could show the role of Alfvén waves for these narrow FAC structures. Their findings nicely complement our results derived for this class of FACs. They are included now in the Discussion section, see responses to Major Item 2).

Line 444: At line 415, delta-t was noted as 16 s, not 18 s. Please confirm and be consistent

Thank you for spotting that. 16 s was a typo which is now corrected.

Line 476: specify the size range where it says "in this size"

Considered

Line 536: how were the quiet days selected for the calculation of the mean merging field and what is the value of this mean?

This part of the Discussion has been greatly revised and the former Figure 10 is no longer included. We have discovered a different source for the deselection of the small-scale FACs, namely the contamination by spectral noise from the very intense kilometer-scale FACs. This modification further enhances the focus on the small-scale FACs.

Line 540-542: Are these results from another paper? The results shown do not show large amplitude FAC structures are prone to instability nor the the FAC current density largely depends on driving.

The Discussion Section 5.2 has largely been rewritten. Our arguments for the deselection of small-scale FACs during times of enhanced solar wind input has changed. We realized that there is a class of even smaller, so-called kilometer-scale FACs which can attain very large amplitudes (see Neubert and Christiansen (2003); Rother et al. (2007)). Although our present dataset is not well suited to cover those km-FACs, our shortest period band (1-3 s) provides a glimpse of the activity in that range. We thus investigated the amplitude ratio $RMS_{1\text{-}3s}$/$RMS_{3\text{-}7s}$ separately for the selected and deselected events within the $d_{cross}$ < 6 km and $\Delta t$ < 18 s range. The resulting distributions of ratios from these event types show two clearly separated functions. From fitted Gauss curves (see below, the new Fig. 10) we obtain for the stable events a mean ratio of 0.65 ±0.22 and for the deselected a somewhat broader distribution with a ratio of 0.93 ±0.28.

[Figure]

Figure . Distribution of the amplitude ratio $RMS_{1-3s}/RMS_{3-7s}$ separately for the selected (left) and deselected (right) events in the $d_{cross} < 6$ km and $\Delta t < 18$ s range. The fitted Gauss curves reveal two clearly separated functions. For the stable events a mean ratio of 0.65 ±0.22 is obtained and for the deselected a broader distribution with a peak at 0.93 ±0.28.

The resulting distributions are in favor of our suggestion that large km-scale FACs can compromise the correlation of the 3-7 s period signals at the two spacecraft. As mentioned by Rother et al. (2007), the intense km-scale FACs tend to come as solitary current spikes (e.g. see Fig. 7, lower frame), thus causing almost a white signal spectrum. The spectral leakage from these spikes will markedly contribute to the 3-7 s period signal, and due to the short life-time of the spikes (order of 1 s) in totally different ways at the two Swarm satellites. It is thus no surprise that the peak cross-correlation coefficient of the 3-7 s signal is reduced when the km-scale FAC amplitudes are large.

For a closer inspection of the $RMS_{1-3s}/RMS_{3-7s}$ ratio distributions we have plotted in the new Figure 11 (see below) how the ratios vary, separately for selected and deselected events, over our study time. The ratios resulting from the selected events stay on an almost straight line at constant level, as derived from the distribution curve in Figure 10, independent of season and local time. The ratio curve for deselected events is more variable but stays all the time above that of the selected. A comparison between the *Em* curve with those for the ratios shows not obvious correlation. There seems to be no direct influence of the solar wind input on the size of the ratios. In spite of that, when looking at the actual *Em* values at the times of selected or deselected events, we find on average slightly larger merging electric fields for deselected case by about 0.2 mV/m, see our old Fig. 10. All this provides further support for our suggestion of disturbing signals from the km-scale FACs. A more detailed investigation of the very small FACs is beyond the scope of this paper and will be the topic of a follow-up study.

[Figure]

Figure. Temporal variation of the amplitude ratios, $RMS_{1-3s}/RMS_{3-7s}$ over the study time. Separate curves are shown for the selected and deselected events. The upleg and downleg arcs cover different local times. For comparison, the merging electric field, Em, represents the amount of solar wind input.

Figure 10 and discussion: It would be useful to include an indication of the range of values at each epoch and whether the higher values for "deselected" events are statistically significant.

Figure 10 is no longer part of the manuscript.

Line 603 - 607: it is not evident from this study that large amplitude currents are unstable, just that large amplitude small-scale magnetic perturbations do not meet the criteria for stable FACs. They may be signatures of wave activity, something which is not examined in this manuscript.

This Conclusion Item 3 has been largely rewritten, following the new finding of spectral leakage from kilometer-scale to our small-scale FAC structures.

Line 625-627: this is speculation with no citations and no discussion in the preceding manuscript, so I suggest removing this sentence.

The sentence has been removed.

---

## Author Comment (AC2)

**Responses to the Reviewer #2 comments on the manuscript [angeo-2024-28]**
"Small- and meso-scale field-aligned auroral current structures, their spatial and temporal characteristics deduced by Swarm constellation"

by Hermann Lühr and Yun-Liang Zhou

We are grateful to the reviewer for their thorough review of our manuscript. Their constructive comments has significantly helped us to improve the study. We have responded to all the comments and made changes wherever we regarded them as appropriate. Below, please find our point−by−point replies. For the convenience of the reviewers, we have first repeated the reviewer's comment and then give our answer in blue text.

General comments:

The manuscript presents a study of small- and meso-scale field-aligned current (FAC) structures using Swarm satellites. The study presents interesting statistical properties of these currents obtained from the close and unique positions of the Swarm constellations. The study finds that merging electric field can affect FAC structures and densities. The statistical properties reported in the manuscript may be useful for researchers of FACs. However, there are serious concerns, which shall be divided into two parts: (a) presentation and (b) technical.

Thank you for the generally positive rating of the study. We have tried our best to improve the manuscript, following the suggestions of the reviewer.

Presentation: The labels on all the figures are difficult to read because they are too small. Please increase the label font sizes on all figures. The manuscript contains many typographical errors, some of them are pointed out below, but the authors should go through the manuscript carefully to check the English grammar and errors. Finally, the organization can be improved. Section 5 (discussion) does not read like a discussion, but rather a continuation of the Section 4 (results/analysis). The discussion section usually provides context, interpretation, physical insights gained from the results (see below).

We have enlarged the labelling of the figures wherever regarded necessary. Limitations on the sizes are imposed by the available space.

We tried our best to remove the errors in English. Many thanks for spotting some of the typos.

Following your suggestion, we added now in the Discussion some paragraphs about magnetospheric processes that are know to generate small-scale FAC at auroral latitudes.

Generally, the study is now more focused on the small-scale FAC structures (10-50 km scale sizes), which have never before been investigated in comparable detail. Please see also our Responses to Reviewer #1.

Technical: The manuscript does not provide much physical interpretation of the results, which usually goes into the discussion. What causes the small- and meso-scale FAC structures? How does merging electric affect the FAC structures on the dayside and more curiously on the nightside? What is the mechanism? What causes the local time variations? It would be nice if the manuscript can discuss some these questions. Perhaps, the following example may help.

In the Discussion we have added an additional subsection on *Possible drivers for the small-scale FACs*

" We may ask, which magnetospheric processes are responsible such filamentary FAC structures. In the literature several suggestions can be found. We may start with the noon time, where the small-scale FACs appear particularly frequent. In this local time sector flux transfer events are believed to be the main source of transient and filamentary FAC structures. They manifest themselves optically as poleward moving auroral forms (e.g., Lookwood et al., 1990; Omidi and Sibeck, 2007). The related field-aligned currents are expected to spatial scales down below 100 km, thus fitting into the 3-13 s period range. On the duskside a viable generation process for transient filamentary FACs is the formation of Kelvin-Helmholtz plasma vortices. They are a result of strong plasma flow sheer in the range of the LLBL. According to Johnson et al. (2021) filamentary FACs in the scale 50-100 km in the ionosphere are expected to connect to the vortex centers at the LLBL. For the example they presented they find a scale of about 70 km. Also this fits into the range of our class of small FACs. A transient phenomenon on the morning side are the travelling convection vortices (TCV) (e.g. Fries-Christensen et al., 1988; Lühr et al., 1998). They are caused by local pressure pulses in the solar wind causing undulations of the magnetosphere that move with the solar wind from the day to the nightside. Related ionospheric effects propagate from the prenoon sector to the morning side. The magnetopause undulations are coupled by a pair of oppositely directed FACs to the ionosphere. Unfortunately, there exist so far no reports of FAC observations by satellite that could be related directly to TCV observation. Even though, filamentary FACs structures with scales of less than 100 km in the ionosphere are also expected from this phenomenon. The dynamic nature of all the mentioned processes infers that they are influenced by kinetic Alfvén waves."

FACs of different spatial scales at different MLTs may be caused by different processes. Generally, large scale FAC structures such as R1 and R2 (Iijima and Potemra, 1976) are fairly stable, but superimposed on these large FAC structures are small and meso scale FAC structures that may be transient in nature. For example, in the afternoon near the open-closed boundary, large scale upward FAC with spatial scale hundreds of km (but can be as small as several tens of km or as large as 1000 km in some cases depending on the solar wind condition) can be attributed to the velocity shear between the solar wind and magnetospheric plasma at the magnetopause boundary layer (Lyons, 1980, Siscoe, 1991, Echim et al., 2008, Johnson and Wing, 2015; Wing and Johnson, 2015). This large scale upward FAC structure is nearly always present because the velocity shear at the magnetopause boundary is always present (but the FAC thickness and strength may vary depending on the solar wind condition). Superimposed on this large scale FAC are small/meso scale FAC structures of tens of km (~50-70 km) in the afternoon sector can be linked to the KH vortices at the magnetopause boundary layer at dusk flank (Petrinec et al., 2022; Johnson et al. 2021). These KH vortices are not static but rather they move anti-sunward with the solar wind and hence the small and meso scale FAC structures associated with these vortices are not static either. Perhaps, a discussion along this line in the introduction and/or discussion section may help the readers appreciate how the statistical results presented in the manuscript may help improve the physical understanding of the magnetosphere and ionosphere and help provide context.

Thank you for the advises. They have been very helpful and guided us to improve our Discussion.

Specific comments:

1. Line 45. "dominate" should be "dominant"
   Done

2. lines 70-71, the sentence is a bit awkward. Would the following capture the meaning better?
The sentence has been revised (line 88-89)
"Larger-scale FACs (>150 km) can be regarded as quasi-stationary, being stable over more than 60 s. The longitudinal extent of the small FAC sheets was reported to be about 4 times large on the nightside than their latitudinal scale, but on the dayside both scales were found to be of comparable size. In spite of these valuable results, the Lühr et al. (2015) study had a number of limitations."

The longitudinal extension of the small FAC sheets on the dayside was found to be comparable to the latitudinal width, but 4 times larger than the latitudinal width on the nightside.

1. line 91, "erose" should be "arose".
Done

2. lines 134-136, What are the assumptions here? Would this technique only work at high latitude, e.g., in the auroral oval? If so, please state.

It is now mentioned that the limitation to the two horizontal components is justified at auroral latitudes since the magnetic field lines are almost vertical to the Earth surface.

3. lines 147-148, do the authors mean FAC intensity (unit = A/m) or density (unit = A/m^2)? FAC density is usually obtained by dividing FAC intensity by the width of the FAC (see for example, Ohtani et al., JGR, 2005).

Thank you for the advice. We use now consistently FAC density.

4. line 184, should FAC density be closer to 15 microA/m^2 (100/7.5) rather than 10 microA/m^2?

No, we think our conversion from $\Delta B$ to FAC density is more correct. According to Eq. (2), $\Delta B$ has to be divided by $\mu_0$ and the satellite velocity.

5. line 240, "base" should be "basis"
Done

6. Figures 3-5 (and other figures), rather than displaying the six period ranges in sec, would it be more useful and practical to display them in spatial scale, i.e., km? Most readers would care more about the spatial scale of FAC structures rather than delta t.

We mention now in the figures both the period band and the scale size. Furthermore, we have added now a table which summarizes all parameters used for the cross-correlation analysis.

| Period band | Scale-length | Data interval | Step size |
|---|---|---|---|
| 1 -3 s | 4-11 km | 32 s | 4 s |
| 3 - 7 s | 11-26 km | 32 s | 4 s |
| 7 - 13 s | 26-49 km | 32 s | 4 s |

| 13 - 23 s | 49-86 km | 64 s | 8 s |
|-----------|----------|------|-----|
| 23 - 39 s | 86-156 km | 64 s | 8 s |
| 39 - 60 s | 156-225 km | 96 s | 12 s |

7. lines 279-280, how do the authors remove the local time and seasonal effects?

The sentence has been rephrased. It is made clearer that our approach normalizes possible dependences of the event occurrence rates on local time and seasonal or other effects.
"When normalizing the identified stable FAC structures by the number of wavy signal events with amplitudes above the threshold (RMS > 2 nT/s), the environmental influences on the occurrence rates of such features, depending possibly on solar wind input, local time or season, are largely removed, but the effect of the Swarm constellation on positive detections prevails. Therefore, these plots are more suitable for evaluating the properties of the detected currents."

8. lines 282-283, "larger ratios of stable FACS are obtained in the afternoon sector". Perhaps, something is missed. Where can we see this in Figure 5? Figure 5 shows that afternoon sector has lower ratios.

We admit that our description was a little bit too short. Now the dependence of stable FAC occurrences on the local time has been outlined in more details for both hemispheres.
" During the first half of the study period we find the blue curves in Figure 5 on higher levels than the red in the northern hemisphere. This means larger ratios of stable FACs are obtained in the afternoon to late evening sector, compared to the early morning to prenoon local times during summer season. Conversely, when looking at the southern hemisphere (right frame), where winter conditions prevail during the first half, the red lines tend to be higher than the blue. This means, a higher percentage of stable FACs in the morning than in the evening sector."

9. lines 310-311, would 10-20 km is more precise for "1-3 s periods "than "10 km"?

We now clarified in Section 2 that the quoted spatial scales amount to halve of the wavelength. Wavelength includes both the pair of up and downward FACs while the scale refers to only one of them.

10. lines 313-319, In Figure 5 there are local time variations as well as temporal (UT) variations. The authors would like to attribute the minimum FAC ratios to 4 Nov 2021 storm. However, the minimum is located in the afternoon sector or morning sectors. Is there an ambiguity in local time vs. UT (storm) effect?

Thank you for pointing that out. We have now commented on the obvious differences of occurrence ratios, at least in the northern hemisphere, between afternoon and morning sectors for the very quiet and stormy days.
" The storm-related reduction of stable FACs on 4 November is present at all wavelength but is more prominent at shorter scales. Conversely, for the very quiet days, 27-29 October 2021, a dip in ratio appears only at long periods, and, at least in the northern winter hemisphere, it is much more evident in the morning than in the evening sector."

11. line 502 and elsewhere in the manuscript, the terms "selected" and "deselected" are a bit confusing. What do the authors mean by "deselected" and "selected". Can these terms be described more clearly?

    We have now introduced a definition for "selected" and "deselected" in Section 5.2. " In the following we use the term "selected" for those current structures in the above defined scale range that passed the stability checks and "deselected" for those not passing the checks."

12. line 538, "cause" should be "caused"
    Corrected

13. Figure 8, what causes the gap near the top right? Is it the result of the constraint on the selected periods? Please explain.

    The blank areas at the top and on the left side are caused by constrains of the satellite constellations. The 2-s along-track separation occurred only during the two weeks close to coplanarity. Similarly, during the period of increased along-track separation (up to 41 s) happened at a time when the longitudinal separation did not exceed 12 km. These constellation constrains are now mentioned in the context of Figure 8.

14. Section 6. The manuscript provides many statistical results, which, at times, are hard to keep track. It would be nice if the authors can provide a table, which can summarize the various properties of the FAC structures.

    Rather than adding a table, we prefer to list the major results verbally in the Conclusions, Section 6.

---

## Author Response (AR2)

**Responses to the Reviewer #1 comments on the revised manuscript [angeo-2024-28]**

"Small- and meso-scale field-aligned auroral current structures, their spatial and temporal characteristics deduced by Swarm constellation"
by Hermann Lühr and Yun-Liang Zhou

We are grateful to the reviewer for the continued effort to review our manuscript. The constructive comments have helped to improve the study. Below, please find our replies. For the convenience of the reviewer, we first repeated the reviewer's comments and then give our answers in blue text. Modifications in the manuscript are marked as red text.

This is my second review of this manuscript. The manuscript seeks to examine field-aligned currents across a range of scales using data from the Swarm spacecraft, notably examining the extent (both spatially and temporally) that these currents are stable by using a period when the spacecraft separation was changing. They provide potentially interesting new insights into small-scale (km scale) currents.

Thank you for the positive rating of our revised manuscript.
Meanwhile we have extended our previous study to the smallest scale FAC structures at auroral latitudes, by making use of the high-resolution 50 Hz magnetic field samples of the Swarm A and C satellites. The close connection between the small-scale FACs (5-40 km scales) and km-scale FACs (0.5-5 km) becomes very evident. In the present study we could, due to limited time resolution, only suggest such a connection, but the new study shows that km-scale FACs only appear when the small-scale FACs exceed a certain amplitude. At the same time, the broad-band signal of the km-FACs contaminates the small-scale FAC B-field variations. All this confirms the conclusions derived in the present study. If there is an interest, a pre-print of the new study is available at:
https://editor.copernicus.org/EGUsphere/ms_records/egusphere-2025-1961

In the revised manuscript, at the end, an editorial note is added to this follow-up study.

The revised manuscript addresses many of the points raised in the earlier review, however I don't think the essence of my concern over the methodology has been fully addressed. The revised manuscript clarifies the data intervals and step sizes used for the different filtering bands but still shows that the number of possible FAC perturbations possible in each window varies for different filter bands. For example, for the 1-3 s band a 32 s data interval is used meaning 32 to ~11 perturbations can occur within that interval. The same interval is used for the 3-7 s band, which means ~11 to ~5 perturbations can occur within that interval. This has the potential to bias any cross correlation. In addition, because the step sizes are also held constant for a given data interval, the step size can varies between just over the size of one perturbation in the 1-3 s band to less than half a perturbation in the 7-13 s band. Given that the comparison between the bands is a key result of this manuscript, I believe it is necessary to be assured that the variation in relative interval and step size is not influencing the results. My suggestion to the authors is to pick use data intervals that are, for example, 3 times the longest period in the filter band and a step size that is half the data interval to test that their results are robust.

We have tested the reviewer's suggestion and repeated the cross-correlation analysis by using a set of time intervals and step sizes that closely follow the period ranges of our six signal bands. This change in processing scheme has hardly any effect on the obtained results. For demonstrating that, below are the original Figure 5 and the newly processed Figure 5 shown, as examples. Just the occurrence rate of the shortest period, 1-3 s, has become slightly larger.

In the revised manuscript we have adapted the proposed set of cross-correlation parameters, as listed in the new Table 1. We admit that it is more plausible. As a consequence, also the Figures 3, 4, 5, 6 and 8 have been replaced with the newly obtained. Luckily, no significant changes have resulted from this reprocessing. Therefore, hardly any changes in the manuscript are required. All the derived conclusions are still valid. Even though, we think, the study approach has become more convincing with this modification

[Figure]

**Figure 5.** The same format as Figure 3, but for the ratio of detected static FAC structures (Fig. 4) divided by all the number of wave structures presented in Figure 3.

[Figure]

**Figure 5. New version**

Minor comments:
Line 23 - I am not sure what is being referred to when the text says "it is weaker". If it is the strength of the small-scale FACs then could I suggest "these FACS are weaker"

The text has been changed accordingly.

Throughout - the text refers to "along track wavelength" (Line 118), "scale size" and "horizontal size" (Line 182) and similar within the text. I suggest it would be helpful to the reader if these were referred to consistently. I would proffer that along-track and cross-track scale size or longitudinal and latitudinal scale size would be a good way to refer to the different scales.

We use the term "along-track wavelength" deliberately in the beginning of the manuscript because it provides a direct relation to the apparent period of the signal recorded by the satellites. This changes from line 176 onward, where we define the scale size, which is halve the wavelength. Subsequently, we refer to FAC scale size or scale length, as recommended. In individual later cases, where the term "wavelength" still appears, it better fits the context. The term, e.g. "horizontal scale" is only used when referring to related statements in cited papers.